# A combination treatment based on drug repurposing demonstrates mutation-agnostic efficacy in pre-clinical retinopathy models

Henri Leinonen [1] ✉, Jianye Zhang[2], Laurence M. Occelli[3], Umair Seemab[1], Elliot H. Choi[2], Luis Felipe L.P. Marinho[3], Janice Querubin[3], Alexander V. Kolesnikov[2], Anna Galinska [4,5], Katarzyna Kordecka[4,5], Thanh Hoang[6], Dominik Lewandowski [2], Timothy T. Lee [2], Elliott E. Einstein[2], David E. Einstein[2], Zhiqian Dong[2], Philip D. Kiser [2,7,8,9], Seth Blackshaw [10,11,12,13,14], Vladimir J. Kefalov[2,7], Marcin Tabaka [4,5], Andrzej Foik [4,5], Simon M. Petersen-Jones[3] & Krzysztof Palczewski [2,7,15,16,17] ✉

Inherited retinopathies are devastating diseases that in most cases lack treatment options. Disease-modifying therapies that mitigate pathophysiology regardless of the underlying genetic lesion are desirable due to the diversity of mutations found in such diseases. We tested a systems pharmacology-based strategy that suppresses intracellular cAMP and Ca2+ activity via G protein-coupled receptor (GPCR) modulation using tamsulosin, metoprolol, and bromocriptine coadministration. The treatment improves cone photoreceptor function and slows degeneration in Pde6βrd10 and RhoP23H/WT retinitis pigmentosa mice. Cone degeneration is modestly mitigated after a 7-month-long drug infusion in PDE6A-/- dogs. The treatment also improves rod pathway function in an Rpe65-/- mouse model of Leber congenital amaurosis but does not protect from cone degeneration. RNA-sequencing analyses indicate improved metabolic function in drug-treated Rpe65-/- and rd10 mice. Our data show that catecholaminergic GPCR drug combinations that modify second messenger levels via multiple receptor actions provide a potential disease-modifying therapy against retinal degeneration.

Most inherited retinal degenerations (IRDs) are currently inaccessible therapeutically, comprising an unmet medical need for a substantial population worldwide. IRDs affect approximately 1 in 2000 people globally[1], and often lead to blindness in childhood[2]. IRDs, including distinct forms of retinitis pigmentosa (RP), are associated with hundreds of distinct genetic mutations (https://sph.uth.edu/retnet/). Management of all those mutations by targeted therapies, mainly gene therapy, is likely to remain impractical and prohibitively expensive for the foreseeable future. To date, only a single gene therapy has been approved for IRDs; namely, a subretinally injectable RPE65 gene-replacement therapy, Luxturna® (voretigene neparvovec)[3]. The proportion of RPE65 gene mutations in IRD cohorts is only ~1%[4]. Although visual function improvement after treatment with Luxturna® has persisted for at least 7.5 years, it is

unclear if it can prevent progression of photoreceptor degeneration[5]. Achieving a therapeutic effect to slow degeneration is in general a major challenge for retinal gene therapies and depends largely on intervention timing and viral distribution.

Alternatively, a disease-modifying treatment (DMT), which aims to substantially diminish pathological mechanisms that drive cells to their demise regardless of the underlying etiology, could be more attainable for larger patient populations, and could benefit more patients faster. DMTs in development often utilize drug repurposing, which can significantly reduce drug development risks, timeline, and costs[6]. This type of treatment could be efficacious not only for IRDs but also for other forms of retinal degeneration (RD).

RD often begins with the rod photoreceptors, with deterioration of dark adaptation and dim light vision[7,8]. The rod photoreceptors comprise a majority of the oxygen-using cells in the outer retina and as these cells die, particularly in RP, the outer retina is exposed to uncontrolled hyperoxia[9]. This oxidative stress leads to damaging effects on bystander retinal cells, particularly the cone photoreceptors. As the cones mediate daytime vision, which is crucial for modern living, the primary goal of RD therapeutics is often cone-protection[9]. The window of therapeutic opportunity for this is often several years[9–11], representing the lag time between rod death and cone death.

Oxidative stress is a major pathophysiological process in RD, which promotes excessive $Ca^{2+}$ influx into the cytoplasm from the extracellular environment and from the endoplasmic reticulum (ER) stores via inositol 1,4,5-trisphosphate ($IP_3$)-mediated release[12]. In turn, increased $[Ca^{2+}]$ in the cytoplasm causes $Ca^{2+}$ influx into mitochondria, which accelerates and disrupts normal metabolism leading to toxicity[12,13]. Since increased intracellular $[Ca^{2+}]$ further accelerates the production of reactive oxygen species[14], a vicious cycle between oxidative stress and elevated $[Ca^{2+}]$ can incur. Oxidative stress can also regulate cyclic adenosine monophosphate (cAMP) signaling, another intracellular second messenger whose increased activity has been associated with RD progression[15]. cAMP is coupled with protein kinase A (PKA) activity, which, among its many signaling processes, can stimulate L-type calcium channels and $Ca^{2+}$ influx[16]. Studies have shown that calcium-channel blockers can delay degeneration in Drosophila[17] and mouse RD models[18]. Cell protective effects have also been demonstrated with drugs that indirectly mitigate intracellular $Ca^{2+}$ and/or cAMP signaling via G protein-coupled receptor (GPCR)-actions[19,20]. GPCRs mediate a plethora of physiological functions and constitute the largest family of druggable targets[21].

Among GPCRs, catecholamine-neurotransmitter receptors are especially highly utilized in pharmacotherapy due to their numerous effects on neuronal signaling. Agonists of the $D_2$-like dopamine receptors (i.e., $D_2/D_3/D_4$ receptors) have shown neuroprotective potential in preclinical settings[22]; however, clinical studies to date have resulted in inconclusive outcomes[22,23]. Adrenergic receptor antagonists, such as $\alpha_1$- and $\beta_1$-blockers, can inhibit NADPH oxidase, leading to diminished generation of superoxide radicals[24,25]. These antagonists exert cardiac cell protection[24,26,27], and potentially retinal neuroprotection[26] in vivo. The $\alpha_2$-agonist brimonidine is a glaucoma drug that displays intraocular pressure-independent therapeutic effects in multiple retinal neurodegeneration paradigms[20]. $\alpha_2$-agonists inhibit cAMP production and calcium channels, decreasing neurotransmitter release in adrenergic neurons[28]. In Phase II trials, intravitreal-implant delivery of brimonidine moderately slowed progression of lesion size in dry age-related macular degeneration-associated geographic atrophy[20,29]. Overall, treating neurodegeneration, including RD, remains a major clinical challenge. It is well-accepted that multitarget therapies may be needed to achieve stronger therapeutic effects in complex diseases such as RD[27,30]. This is attributable to the flexibility and redundancy of biological systems, particularly in the nervous system, that allows them to compensate if just a single element is targeted[27].

We have developed a systems pharmacology-based DMT that utilizes three simultaneously administered GPCR drugs[19,31,32], which reach the eye after systemic administration[32] and which target receptors that are expressed in both mouse and human retinas[19]. Tamsulosin (T) blocks the $G_q$-linked $\alpha_1$-receptors; Metoprolol (M) blocks the $G_s$-linked $\beta_1$-receptors; and Bromocriptine (B) activates the $G_i$-linked D2-like dopamine receptors. Mechanistically, $G_q$-receptor-antagonism decreases the $IP_3$-mediated release of $Ca^{2+}$ ions from the endoplasmic reticulum (ER), while both $G_s$-receptor-antagonism and $G_i$-receptor-agonism lead to decreased adenylyl-cyclase (AC) activity and turnover of intracellular cAMP. We hypothesize that suppressive effects on these second messenger pathways protects against RD[32]. Indeed, combined "TMB treatment" at low doses of each compound prevents acute bright light-induced RD in mouse models of Stargardt disease[19,32], Oguchi disease[31], congenital stationary night blindness[31], and in wild-type (WT) albino mice[19]. The required doses for a therapeutic effect against light-induced RD are much larger if the same drugs are administered individually as a monotherapy, suggesting therapeutic synergy by the TMB drug cocktail[19]. However, the prior evidence of efficacy is considered preliminary, based solely on acute administration. Many acutely beneficial treatments can in fact be harmful in prolonged use[33]. The current study provides the first highly translationally relevant proof-of-concept for the efficacy of TMB treatment during chronic administration in multiple types of IRD models.

## Results

### Dietary TMB treatment attenuates the aberrant retinal transcriptome, slows retinal degeneration, and improves visual function in the rd10 mouse model of autosomal recessive RP

Retinal protection by the TMB drug combination was demonstrated in an acute light-induced RD paradigm, wherein drugs are administered prophylactically prior to damage induction[19,31,32]. Moving towards clinical translation necessitates comprehensive evaluation of drug efficacy after prolonged administration in chronic animal models that mirror human disease more directly. Furthermore, as the goal of a DMT is to decrease pathology in an etiology- or mutation-agnostic manner, we needed to test the efficacy of TMB treatment in several distinct disease paradigms where the time-course and mechanisms of the RD differ.

To begin exploratory chronic drug trials in mouse models (Fig. 1A; Supplementary Data 1), we needed to establish a suitable method of drug administration. Based on our prior study, T, M, and B are quickly eliminated from circulation after an intraperitoneal administration in mice[32], suggesting a short half-life of elimination. Accordingly, the drugs were incorporated into standard mouse chow (serving as vehicle, abbreviated in figures as "veh") as follows: T, 0.05 mg per g of standard chow (0.05 mg/g); M, 2.5 mg/g; and B, 0.25 mg/g. As the distinct mouse models used in the study all were based on the inbred C57BL/6 J mouse-strain background, we used young adult wild-type (WT) male C57BL/6 J mice for testing drug concentrations in the serum and target tissues. After three full days of acclimatization to the diets, samples of serum, retina, and eye cup were collected. Drug levels were assayed using LC/MS (Fig. 1B; Supplementary Data 2). In samples collected in the middle of the active period (12 a.m.), the doses of T and M in the serum were $12.8 \pm 2.33$ ng/ml and $2361 \pm 614$ ng/ml (mean ± SD), respectively. The concentrations of T and M in eye cups were much higher than those in the serum, suggesting melanin-binding and drug retention in the melanin-rich eye[34,35]. Detection of T and M levels in serum was also used as a readout to test the utility and uniformity of the dietary drug-administration strategy. Additional experiments with serum samples collected at the end phase of the active period (6 a.m.; T, $13.1 \pm 3.7$ ng/ml; M, $759 \pm 644$ ng/ml) and in the middle of the inactive period (12 p.m.; T, $9.84 \pm 8.1$ ng/ml; M, $803 \pm 761$ ng/ml) indicated constant drug exposure (Supplementary Fig. 1A); and the levels of

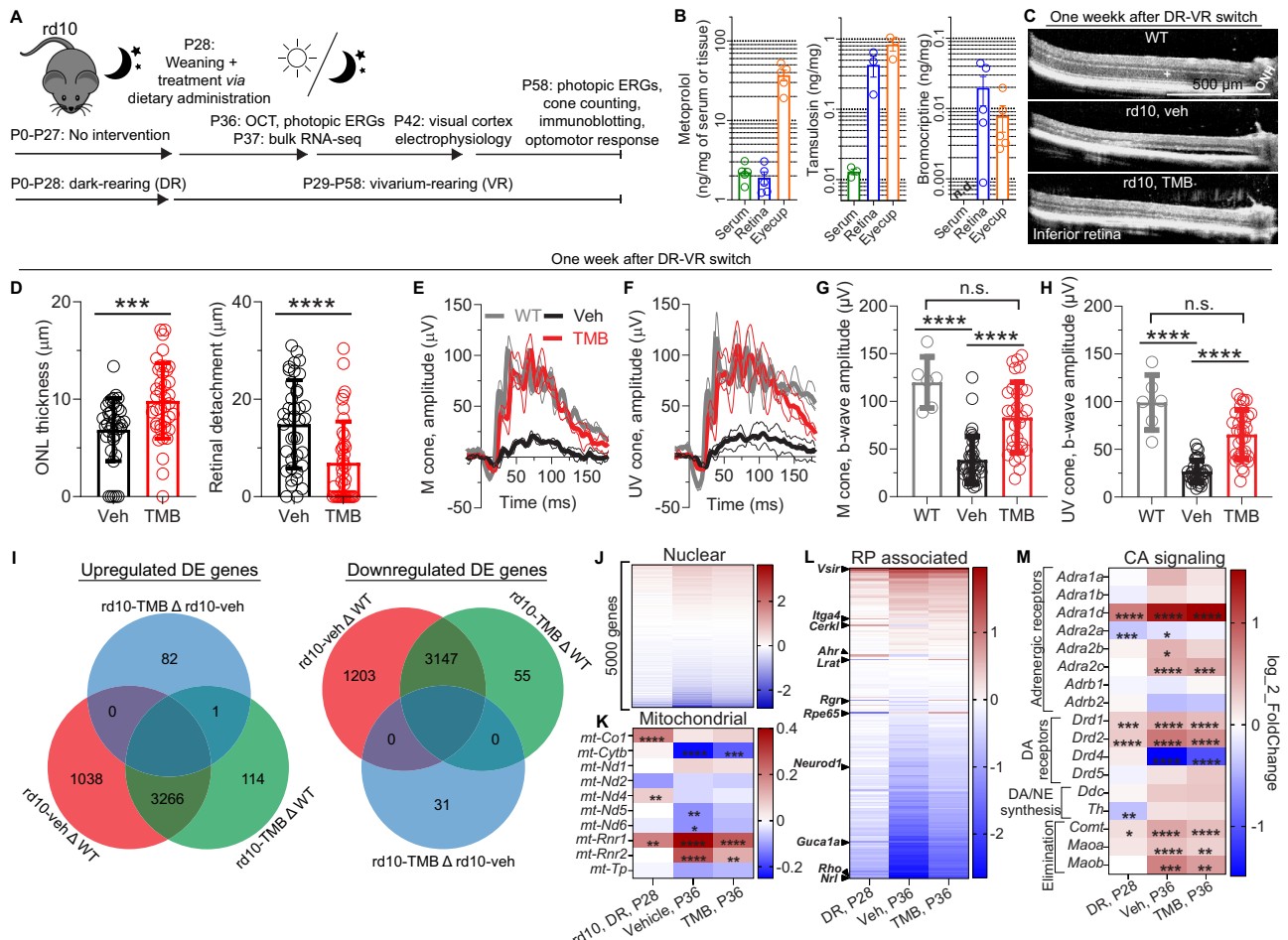

**Fig. 1 | Dietary TMB self-administration stabilizes the retinal transcriptome and mitigates the disease phenotype in rd10 mice. A** Study design. **B** Representative examples of drug doses via dietary tamsulosin, metoprolol, and bromocriptine (TMB) coadministration, obtained from the blood and the target tissues at 12:00 a.m. The blood level of bromocriptine did not reach the sensitivity limit of the assay, so it is labeled as "not detected" (n.d.). **C** Representative optical coherence tomography (OCT) images. The plus sign (+) indicates outer nuclear layer (ONL). ONH, optic nerve head. **D** ONL thickness and retinal detachment, as measured from OCT images. **E, F** Group-averaged electroretinogram (ERG) waveforms in response to green **E** and UV **F** stimuli under photopic conditions in a representative cohort. **G, H** M-cone- **G** and S(UV)-cone-dominant **H** ERG b-wave amplitudes. **I** Venn diagrams of upregulated and downregulated genes in retinal bulk RNA-sequencing. **J, K** Transcriptome heatmap of 5000 most-abundant nuclear genes **J**, and 10 most-

abundant mitochondrial-encoded genes **K**, that show an expression change compared to WT in dark-reared (DR) rd10 mice (at P28, non-treated), or experimental rd10 mice (P37, changed from dark to vivarium/cyclic light-rearing [CLR] at P29) that were treated with vehicle (veh) or TMB. The asterisks signify statistical differences (adjusted $P < 0.05$) compared to WT expression level. **L** Heatmap of regulation in a set of genes associated with retinitis pigmentosa (RP). **M** Heatmap of regulation in genes encoding adrenergic and dopamine receptors, and major catecholamine synthetizing or degrading enzymes. The statistical analysis employed for graphs in **D** was the Mann–Whitney U-test (two-tailed), and for graphs **G** and **H** the Kruskal–Wallis test was followed by Dunn´s multiple comparisons test; The asterisks signify: ***$P < 0.001$, ****$P < 0.0001$. Bar graph data are presented as mean ± SD. Source data are provided as a Source Data file.

these drugs remained at a high range relative to clinical doses derived from literature reports[36,37]. Published clinical pharmacokinetic (PK) data suggest up to 18 ng/ml ($C_{max}$) serum level for T[36], and 340 ng/ml ($C_{max}$) for M[37] when acute oral administration was used with relevant doses for the treatment of benign prostate hyperplasia (T) and cardiovascular indications (M). While we could detect chromatographic peaks attributable to B in serum samples, the sensitivity of our method did not always reliably allow its quantification. Clinical PK data suggests that B serum levels during continuous treatment of Parkinson's disease can be relatively low; *i.e.*, <5 ng/ml[38]. However, we measured consistent B concentrations above the detection limit in the samples of retina and eye cup (Fig. 1B). Throughout the studies, we also followed body weight gain as a sensitive marker of appetite and general animal wellbeing and found no distinction between animals on control/vehicle-diet *versus* the TMB treatment (Supplementary Fig. 1B–D). Overall, dietary TMB administration proved to be applicable for drug-efficacy testing in chronic mouse RD paradigms, although serum level variance

with the method is large and the drug levels tested at single time points in mice are not directly comparable to typical exposure patterns in humans.

We started drug-efficacy trials with the $Pde6\beta^{rd10}$ mouse model (known as the rd10 mouse), representing a rapidly progressing RD. The majority of the rods die by postnatal day P30 if the rd10 mice are reared under standard light conditions[39]. As we could start treatments only after weaning the mouse pups, we preferred slowing the disease progression by dark-rearing the rd10 mice until initiation of TMB treatment at P28. The mice were acclimatized to the new pellets for one day and then transferred at P29 to vivarium rearing/cyclic light rearing (CLR, lights on 6:30 a.m. and lights off 6:30 p.m.). The switch from dark-rearing to vivarium rearing (DR-CLR) induced a rapidly progressing RD and a dramatic loss of the outer nuclear layer (ONL, photoreceptor nuclei loci) of the central retina, and retinal detachment in the vehicle-treated rd10 mice already after seven days in the vivarium, as was evidenced by optical coherence tomography (OCT)

imaging (Fig. 1C). ONL thickness was improved by 43% on average by the TMB treatment (Fig. 1D) and on the other hand alleviated retina´s displacement/detachment from the RPE: inter retina-RPE gap decreased by more than 2-fold at central retinal location (Fig. 1D). Photopic electroretinograms (ERG) in response to monochromatic green and UV flashes were recorded immediately after OCT imaging to assess middle wavelength-sensitive (M-cones) and short wavelength-sensitive (S[UV]-cones) cone-mediated retinal function, respectively. Both M-cone- and S(UV)-cone-mediated responses in vehicle-treated rd10 mice were diminished ~3-fold compared to the corresponding responses of WT mice (Fig. 1E–H). The response waveforms and amplitudes for the TMB-treated rd10 mice were remarkably well maintained, and they were on average more than double in amplitude compared to the vehicle group (Fig. 1E–H). These results suggest that TMB treatment is particularly effective at preventing cone-mediated retinal dysfunction in RP pathology. Since the effect of TMB treatment on maintaining the photopic ERG responses of rd10 mice exceeded our expectations, we also treated C57BL/6 J WT mice for 1-month and recorded their photopic ERGs as a control experiment. We found that chronic TMB administration did not alter cone function directly, based on a lack of TMB-induced change in the photopic ERG responses in the WT mice (Supplementary Fig. 1E, F).

Even though we utilized approved drugs whose pharmacodynamics are known for their typical clinical indications, it remains speculative how they mediate retinal protective effects. To provide mechanistic insights that could explain their therapeutic efficacy against RD, we used the same treatment paradigm and collected retinas from vehicle- and TMB-treated rd10 mice ($n = 8$ per treatment, sexes balanced) at P37 for RNA-sequencing (RNA-seq). Our strategy for data analysis was to first define pathological alterations in the rd10 mouse-retinal transcriptome compared to the healthy WT state and then examine whether TMB treatment normalizes the rd10 mouse transcriptome toward the WT profile. Therefore, in addition to the experimental vivarium-reared rd10 mice, we also performed RNA-seq with retinas from age- and sex-matched WT mice ($n = 4$, primary control group), as well as with retinas from fully dark-reared rd10 mice at P28 ($n = 4$, secondary control group). These dark-reared rd10 mice present an early-stage RP phenotype at the time when treatments were to be started. The RNA-seq analysis revealed a remarkable TMB-mediated prevention of the aberrant transcriptome of the rd10 mice (Fig. 1I). Whereas over two thousand differentially expressed (DE) genes (1038 upregulated; 1203 downregulated) were identified in the comparison of the transcriptomes of vehicle-treated rd10 *versus* WT mice, the corresponding numbers were much lower (114 and 55, respectively) for the comparison of the transcriptomes of TMB-treated rd10 *versus* WT. A direct comparison of the transcriptomes of vehicle-treated *versus* TMB-treated rd10 mice yielded a relatively low number of DE genes (Fig. 1I) and did not readily identify transcriptomic changes that would help to explain the treatment efficacy (Venn diagram components shown in Supplementary Data 3). For example, the TMB *versus* vehicle comparison revealed increased expression of several photoreceptor-enriched genes such as *Opn1sw*, *Crx* and *Ush2a*. Such increases are consistent with decreased photoreceptor degeneration as the result of the TMB treatment. Another prominent finding in this comparison was decreased expression of the somatostatin gene *Sst*, which is under adrenergic control[40].

To inspect mitochondrial and nuclear-encoded genes separately, these fractions were processed separately in the RNA-seq analysis. We constructed a heatmap of the 5000 most highly expressed nuclear genes (Supplementary Data 4), showing fold-change relative to WT level (Fig. 1J). Like the Venn diagrams (Fig. 1I), the heatmap indicates stabilization of the transcriptome by TMB treatment, closer to healthy levels (Fig. 1J). Only 10 mitochondrial genes showed sufficient read-count levels for quantitative analysis (Supplementary Data 5). An expression heatmap including all these genes is shown in Fig. 1K, with

asterisks highlighting significant DE genes compared to WT. The same mitochondrially encoded genes showed dysregulation in vehicle- and TMB-treated rd10 mouse groups; however, fold-changes were overall smaller for the TMB-treatment group (Fig. 1K). Next, we investigated the expression of 198 genes closely associated with RP (Supplementary Data 6). Many of these genes showed large fold-changes in rd10 mice compared to WT, but this dysregulation was mitigated by the TMB treatment (Fig. 1L). Finally, we investigated changes in genes encoding catecholamine neurotransmitter receptors, as well as catecholamine neurotransmitter-synthetizing and -eliminating enzymes. Many of the catecholamine neurotransmitter signaling-related genes were significantly upregulated in rd10 mice regardless of TMB treatment (Fig. 1M). *Maoa*, *Maob*, and *Comt* genes that encode enzymes that degrade norepinephrine and dopamine displayed little to no change in the dark-reared rd10 group but showed upregulation in the vivarium-reared rd10 mice.

In a subset of experiments with vivarium-reared rd10 mice, we extended trials to 2-week or 1-month TMB-treatment duration to test primary visual cortex (V1) function or cone-photoreceptor survival, respectively. Both visual evoked potential (VEP) and single-cell recordings demonstrated better-preserved function at the V1 (Fig. 2A–D). The V1 neurons responded stronger to light stimulation in TMB-treated rd10 mice as shown by increased VEP amplitudes (Fig. 2B) and single neurons´ spiking rates (Fig. 2D; vehicle-group = 8.69 ± 12.53 and TMB-group 11.92 ± 16.05 spikes per sec). On the other hand, excessive neural noise during light OFF was mitigated by the TMB treatment as demonstrated by lower background firing rate in TMB-treated *versus* vehicle-treated P42 rd10 mice (Fig. 2D–Q; vehicle-group = 4.31 ± 8.76 and TMB-group 3.85 ± 7.94 spikes per sec). In terminal experiments at P58, immunohistochemistry (IHC) using anti-M- and anti-S(UV)-cone antibody staining in retinal flat mounts showed clearly visible decline in cone density in the central retina for the vivarium-reared rd10 mice at P58 (Fig. 2E–G). Semi-automated cone counting suggested that TMB treatment significantly diminished degeneration of M-cones (Fig. 2H), but for the S(UV)-cones the effect was not significant (Fig. 2I). In contrast, photopic ERG responses for the TMB-treated P58 animals were twice as large as those for untreated animals in both M-cone- (Fig. 2J) and S(UV)-cone-mediated (Fig. 2K) pathways. As UV-light also stimulates the M-cones, the rescue of the M-cones may explain the discrepancy between cone counting and photopic ERG results. Immunoblot analyzes of cone arrestin (ARR3), rhodopsin (RHO) and manganese superoxide dismutase (SOD2) from retinal homogenates corroborated the observation of decreased photoreceptor degeneration after TMB treatment (Fig. 2L) and indicated increased antioxidant defense as a contributing mechanism (Supplementary Fig. 2). We also tested a subset of mice with an optomotor-response (OMR) behavioral-vision paradigm[41], where mice were stimulated with vertical gratings of altering spatial frequency drifting through their visual field. TMB-treated rd10 mice displayed more frequent head movements (higher OMR index; Fig. 2M) in response to narrower stimuli compared to vehicle-treated rd10 mice, corresponding to their 20% higher visual acuity (Fig. 2M insert).

Finally, the drug levels in the serum observed with the "standard TMB diet" (T, 0.05 mg/g, M, 2.5 mg/g, and B, 0.25 mg/g) were relatively high for T and M (Fig. 1B and Supplementary Fig. 1A), so we evaluated a decreased TMB dose. Accordingly, a new pellet batch ("TMB-low diet") with a 5-fold lower dose (T, 0.01 mg/g; M, 0.5 mg/g; and B, 0.05 mg/g) was tested. As expected, this new regimen led to lower levels of the drugs in the serum (Supplementary Fig. 3A) compared to the standard diet (Fig. 1B and Supplementary Fig. 1A). After 1-week of treatment, rd10 mice maintained on the TMB-low diet showed double stronger photopic ERG responses compared to the vehicle group (Supplementary Fig. 3B) and decreased retinal detachment (Supplementary Fig. 3C). Treatment effect on the ERG response was maintained when re-evaluated, 1-month after the treatment onset (Supplementary

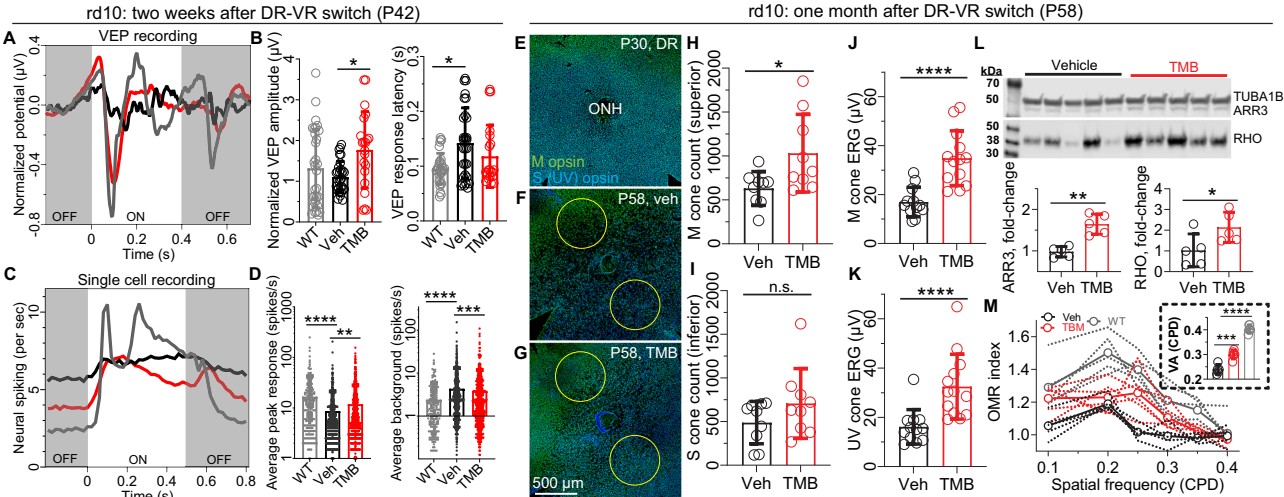

**Fig. 2 | Chronic TMB administration improves visual function and cone viability in rd10 mice.** Study design is depicted in Fig. 1A. This figure presents data after a two-week **A–D** or one-month-long **E–M** treatment period. **A** Group-averaged visual evoked potential (VEP) responses (WT, $n = 7$ mice, $n = 21$ electrodes; rd10-vehicle, $n = 10$ mice, $n = 30$ electrodes; rd10-TMB, $n = 10$ mice, $n = 31$ electrodes). The traces represent group´s mean responses. **B** Normalized VEP amplitudes and first negative component latencies from response to light onset. **C** Group-mean neural spiking rate from single V1 neurons in response to ON-OFF light stimulus. The total number of single cells recorded in WT mice was 217, in rd10-vehicle 638, and in rd10-TMB mice 642 cells. **D** Spiking during ON response peak (left), and background spiking during sustained light OFF (right). **E** Representative partial retinal whole mount, stained for cone opsins, of a dark-reared P30 rd10 mouse. Image is zoomed and centered at the optic nerve head (ONH). **F–G** Comparison of ONH-centered zoomed images of vehicle-treated (**F**) and TMB-treated (**G**) experimental rd10 mice. The circular cone counting windows (yellow, 600 μm diameter) were centered at 500 μm from the ONH. **H** Superior central M-cone and (**I**) inferior central S(UV)-cone counts. **J** M-cone and (**K**) S(UV)-cone ERG b-wave amplitudes. **L** Western blots of alpha-tubulin (TUBA1B, loading control), cone arrestin (ARR3), and rhodopsin (RHO). The graph shows expression differences between the vehicle and TMB groups. **M** Optomotor responses as a function of changing spatial-frequency stimuli. The insert shows extrapolated visual acuity from the OMR index *versus* spatial frequency graphs. The statistical analysis employed for graphs **I**, and **K** was the Mann–Whitney U-test (two-tailed); for graphs (**B**) and (**D**) the Kruskal-Wallis test was followed by Dunn´s multiple comparisons test; for graph M (insert) one-way ANOVA analysis was followed by Bonferroni posthoc tests; and for graphs (**H, J**) and (**L**), Welch´s t-test (two-tailed) was used. The asterisks signify: *$P < 0.05$, **$P < 0.01$, ***$P < 0.001$, ****$P < 0.0001$. Bar graph data are presented as mean ± SD. Source data are provided as a Source Data file.

Fig. 3D). However, while the therapeutic effect on cone function was similarly strong with the standard TMB diet and TMB-low diet after 1-week on treatments (Supplementary Fig. 4A), the superiority of the higher standard dose emerged after 1-month on treatments as the M-cone-mediated ERG responses were over 50% better preserved with the higher TMB dose compared to the low dose (Supplementary Fig. 4B).

### TMB treatment still exhibits efficacy in dark-reared rd10 mice with slower disease progression

Dark-rearing modulates phenotype and delays disease progression in rd10 mice[42,43]. This approach provides an opportunity to test TMB drug efficacy with an alternative paradigm and enables longer-term trials. Accordingly, we housed vehicle- and TMB-treated rd10 mice in a dark room until 4 months of age (Fig. 3A). The only light exposure experienced by these mice throughout their lifespan was dim red light during daily husbandry, brief exposure to monitor-light during monthly OCT imaging, and a few brief green-light flashes during ERG recordings. Treatments were started at P28, and subsets of the animals were tested at intervals to monitor disease progression and to perform mechanistic investigations.

After 1-month on treatment, TMB-treated rd10 mice showed 40% higher scotopic ERG a-wave responses and 30% higher b-wave responses compared to the vehicle group (Fig. 3B–D); but ONL thickness was not different between groups (Fig. 3E–G). It should be noted, however, that retinal macrostructure and ONL layer was relatively well remained in 2-month-old dark-reared rd10 mice (Fig. 3E, F), rendering ONL thickness as an insensitive measure of treatment in this context. Despite early degeneration stage, our analysis using enzyme-linked immunosorbent assay (ELISA) showed doubled whole retinal 4-hydroxynonenal (4-HNE) contents, a standard lipid peroxidation marker, in rd10 mice that were kept on vehicle diet (Fig. 3H). Retinal

4-HNE content in TMB-treated rd10 mice was significantly decreased compared to vehicle-treated rd10 mice (Fig. 3H) and did not significantly differ from that of WT controls. As a supplemental assay to corroborate the finding, we performed immunoblotting to detect catalase (CAT) expression in retinal homogenates from the rd10 mice and found CAT expression increased to a similar extent as 4-HNE content in rd10 mice (Fig. 3I; full blots shown in Supplementary Fig. 5A–C, H). We also performed immunoblotting to detect glial fibrillary acidic protein (GFAP), a typical inflammation marker. GFAP expression was increased in rd10 mice around 6-fold compared to WT mice, but there was no treatment effect on its expression (Fig. 3J, K). As *Comt* was significantly overexpressed in the RNA-seq data (Fig. 1M) and could indicate an adaptive response to increased catecholaminergic activity, we also tested COMT expression *via* immunoblotting. COMT content was elevated in vehicle-treated rd10 mice by around 50% compared to WT values, and to a lesser (not significant) extent in TMB-treated rd10 mice (Supplementary Fig. 5L). SOD2 levels at this stage were unaltered (Supplementary Fig. 5F).

Encouraged by the findings at the earlier mouse ages, we continued dietary TMB administration for 3 months duration in several cohorts of the dark-reared female rd10 mice. IHC of retinal flat mounts revealed a well-preserved cone population with relatively minor density deterioration in the central retina in these 4-month-old dark-reared rd10 mice (Fig. 3L, M; or see larger size images in Supplementary Fig. 6). We sampled M-cone populations at three distinct distances from the optic nerve head (ONH) border in the superior retina, and analogous S(UV)-cone populations in the inferior retina (Supplementary Fig. 6). The overall M-cone counts in the centermost retina (0.3 mm from the ONH border) were 50% higher in the TMB-treated rd10 mice compared to the vehicle group (Fig. 3N) whereas there was no apparent treatment effect on the S(UV)-cone populations (Fig. 3O). Although photopic ERG responses were on average

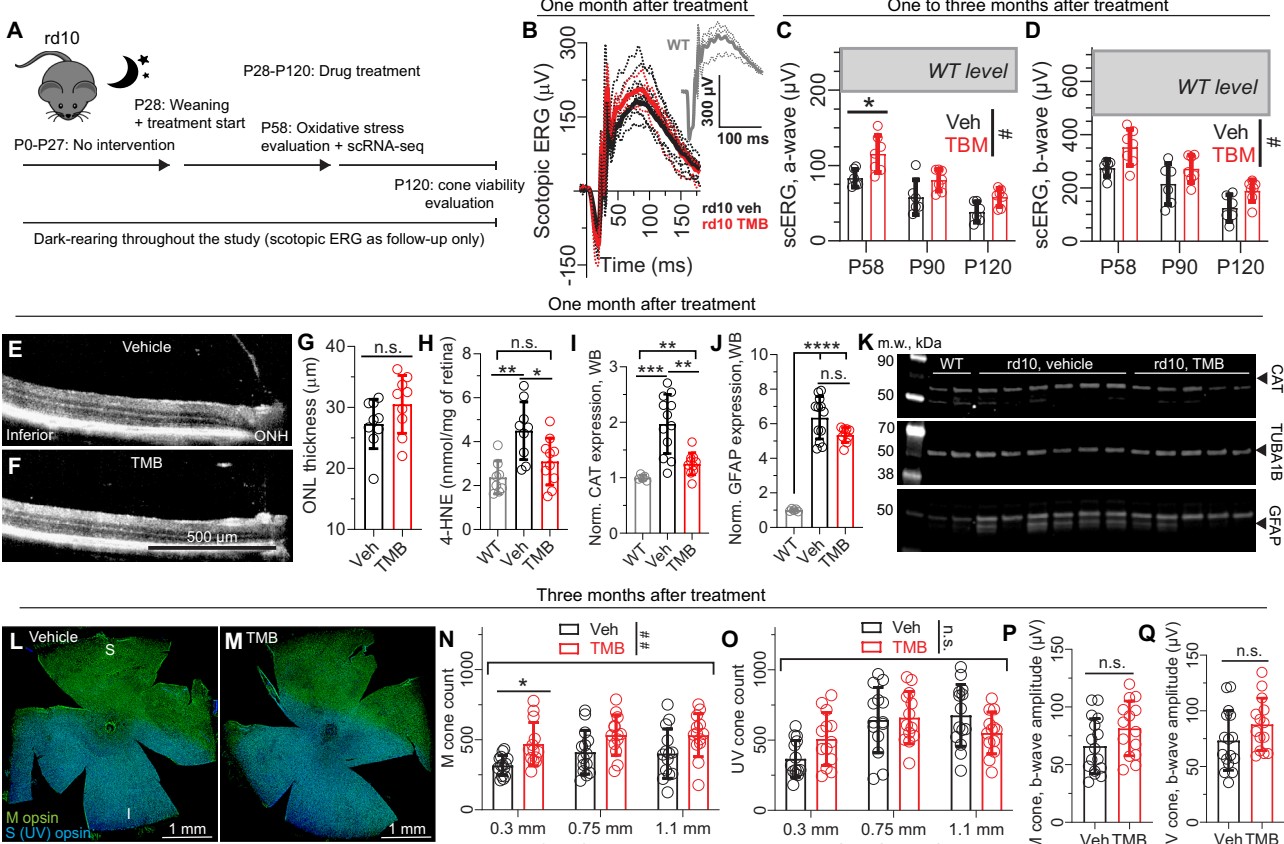

**Fig. 3 | Retinal protection in dark-reared rd10 mice is associated with decreased lipid peroxidation. A** Study design. Note, graphs B and E-K represent data for mice after one month on treatments; graphs **L–Q** represent data for mice after three months on treatments. **B** Group-averaged scotopic ERG waveforms in a representative cohort. **C, D** Scotopic ERG a-wave **C** and b-wave **D** amplitudes at different stages of disease progression in the same mice (repeated measures design). **E, F** Representative OCT images. **G** ONL-thickness assessment. **H** Whole retinal 4-HNE contents, as measured by ELISA (WT, n = 8; rd10-vehicle, n = 9; rd10-TMB, n = 10). **I–J** Quantification of expression of catalase (CAT) and GFAP from immunoblots (WT, n = 5; rd10-vehicle, n = 12; rd10-TMB, n = 9). **K** Representative immunoblots from cohort 1. A total of two cohorts (individual experiments) were performed and the data pooled for the analyzes shown in panels (**I, J**). **L, M** Representative retinal flat mounts stained with antibodies against M-cone

opsin or S(UV)-cone opsin. **N, O** M-cone and S(UV)-cone counts from retinal flat mounts. **P, Q** M-cone- and S(UV)-cone-dominant photopic ERG amplitudes. The three-month-long experiments were performed in female rd10 mice. Welch´s t-test (two-tailed) was used to analyze data in **G, P** and **Q**. ERG data in (**C, D**) (age as within-subjects and treatment as between-subjects factor) and cone count data in (**N, O**) (retinal location as within-subjects and treatment as between-subjects factor) were analyzed by two-way RM ANOVA with the Geisser-Greenhouse correction. The statistical analysis performed for **H** was one-way ANOVA and Bonferroni´s post hoc test, whereas **I, J** were analyzed by Welch´s ANOVA followed by Dunnett´s T3 tests. The pound signs indicate an ANOVA-significant main effect between treatments: #P < 0.05, ##P < 0.01. The asterisks mark significant post hoc test effects: *P < 0.05, **P < 0.01, ***P < 0.001, **** P < 0.0001. Data are presented as mean ± SD. Source data are provided as a Source Data file.

20% higher for the TMB group, the effects were not statistically significant (Fig. 3P, Q).

## TMB-treatment-dependent transcriptomic changes in female rd10 mice, particularly in Müller glia, cones, and cone bipolar cells

Focusing again on earlier stage disease progression, we performed single-cell RNA-seq (scRNA-seq) analysis on retinas from 2-month-old vehicle- and TMB-treated dark-reared rd10 female mice, and on retinas from WT controls. The largest cell cluster identified was the Müller glia (MG) (Fig. 4A). These cells from rd10 mice displayed extensive transcriptomic regulation compared to those from healthy WT mice (Fig. 4B; Supplementary Fig. 7). Evidence of regulation in the rod cluster was moderate, despite the presence of dying rods. Even rod bipolar cells (RBCs) showed more DE genes in the comparison of those from WT *versus* those from the rd10 vehicle group (109 upregulated, 162 downregulated in rd10; Fig. 4B), despite the RBC cluster being half as large as the rod cluster, making statistical testing less robust. The effect of TMB treatment (WT *versus* rd10-TMB) was shown as a more

than 3-fold diminution in the number of DE genes in the RBCs (30 upregulated, 38 downregulated; Fig. 4B) relative to vehicle treatment. Supplementary Data 7–8 show full data tables of WT *versus* rd10-vehicle and WT *versus* rd10-TMB comparisons.

We focused our investigation of TMB efficacy on the genes that were up- or down-regulated by TMB treatment in the MGs of the rd10 mice. Among 279 downregulated DE genes (Fig. 4C), the largest fold-changes (more than 20% downregulation) were observed in genes associated with cellular response to stress, including *Lcn2, Gfap, C4b, Mt1, Mt2, Atf3, Nurp1* and *Cd44* (Supplementary Data 9). 204 genes were upregulated by TMB in the retinal MGs of the rd10 mice (Fig. 4D). More than 20% upregulation was observed for the following genes: *Dbp, Shisal2b, Ciart, Id1, Col9a1, Mif, Hlf, Cav1, Per3, Zfp467, Gngt1, Id3, and Rcn2*. The functional enrichment pattern created by the upregulated DE genes was notably different from that of the downregulated DE genes (Fig. 4C, D).

Focusing on cone survival, we examined clusters 3 (largest CBC cluster) and cluster 4 (cones). Figure 4E & F display all the DE genes (rd10-vehicle *versus* rd10-TMB), in clusters 3 and 4, respectively, and

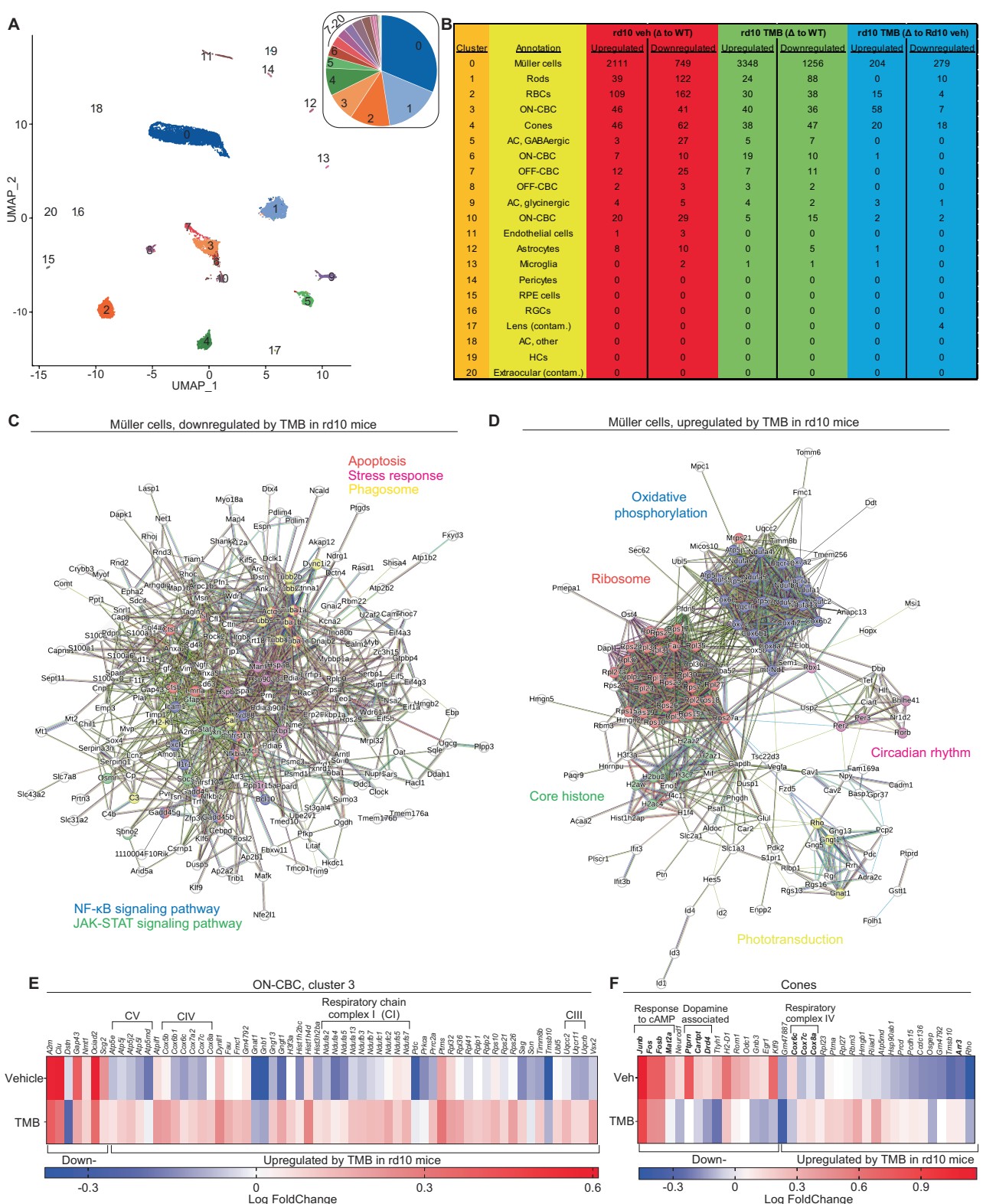

**Fig. 4 | Single-cell RNA sequencing reveals a decreased cell stress- and an improved metabolic status in Müller glia by chronic TMB therapy in dark-reared female rd10 mice. A** Group-consolidated UMAPs by cluster. The insert shows cluster size breakout as a pie chart. Single-cell suspensions were prepared from three rd10 mice per group, whereas the WT group had four mice. Analysis was performed from 4478 cells from WT mice, 5952 cells from vehicle-treated rd10, and 4812 cells from TMB-treated rd10 female mice. **B** Table showing the number of differentially expressed (DE) genes for all clusters and for all groupwise comparisons. The cutoff for DE genes was set at $P < 0.05$ after adjustment for multiple comparisons. **C, D** Gene interaction network (generated by String database version 12.0) of downregulated (**C**) and upregulated (**D**) DE genes in vehicle-treated rd10 versus TMB-treated rd10 mice in the Müller cell(0) cluster. **E, F** DE genes in vehicle-treated rd10 versus TMB-treated rd10 mice in the ON-CBC cluster 3 (**E**) and cone cluster 4 (**F**), shown as heatmaps where regulation is compared to WT-gene expression. AC amacrine cell, HC horizontal cell, ON-CBC ON cone bipolar cell, OFF-CBC, OFF cone bipolar cell, RBC rod bipolar cell, RGC retinal ganglion cell, RPE retinal pigment epithelium.

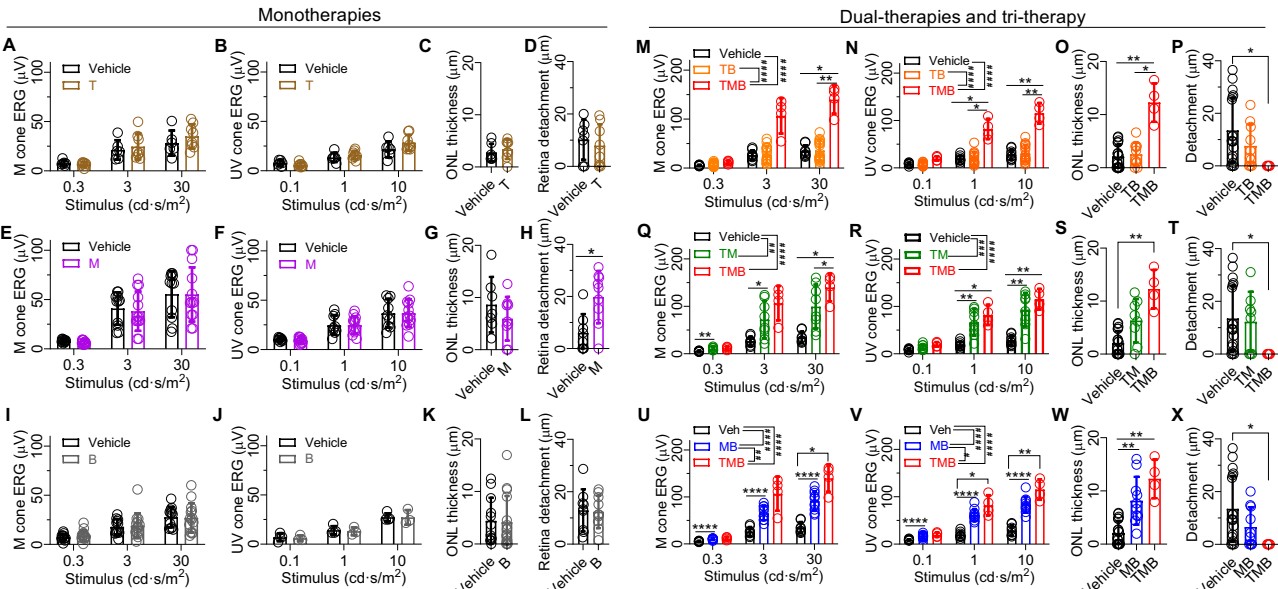

**Fig. 5 | Monotherapies with tamsulosin, metoprolol, or bromocriptine are not effective in delaying retinal degeneration in rd10 mice.** Rd10 mice were dark-reared between P0-P28, and treatments were started at P28. One day later (P29), mice were transferred to normal laboratory housing conditions, and treatment effect was tested at P36. The drug effect was evaluated using four different parameters: photopic M- or S(UV)-cone targeted ERG b-wave amplitudes, ONL thickness, or retinal detachment from the RPE. **A–D** Tamsulosin (T) monotherapy data. **E–H** Metoprolol (M) monotherapy data. **I–L** Bromocriptine (B) monotherapy data. **M–P** Tamsulosin/bromocriptine (TB) dual-treatment data contrasted with TMB triple-treatment. **Q–T** Tamsulosin and metoprolol (TM) dual-treatment data contrasted with TMB triple-treatment. **U–X** Metoprolol and bromocriptine (MB) dual-treatment data contrasted with TMB triple-treatment. The ONL thickness and retinal detachment data were statistically analyzed by the non-parametric Mann–Whitney U-test (two comparisons; two-tailed) or the Kruskal-Wallis test (three comparisons) followed by Dunn´s multiple comparisons tests. All ERG data were analyzed using repeated measures two-way ANOVA with Geisser-Greenhouse correction and followed by Bonferroni post hoc tests. Asterisks illustrate significant Mann-Whitney or Bonferroni test results: *$P < 0.05$, **$P < 0.01$, ***$P < 0.001$, ****$P < 0.0001$. The pound signs illustrate significant between-subjects ANOVA main effects: #$P < 0.05$, ##$P < 0.01$, ###$P < 0.001$, ####$P < 0.0001$. Data are presented as mean ± SD. Source data are provided as a Source Data file.

their regulation heatmap relative to the WT transcriptome. With TMB treatment in rd10 mice, 7 genes were downregulated, and 58 genes were upregulated in cluster 3. The upregulation was largely associated with genes involved in oxidative phosphorylation (Fig. 4E), as with the MGs (Fig. 4D). In cluster 4, 18 genes were downregulated, and 20 genes were upregulated. The downregulation involved small pools of genes associated with cellular response to cAMP and dopamine signaling (Fig. 4F). In line with findings from cluster 0 and cluster 3, several genes associated with cytochrome c oxidase were upregulated in cluster 4 by TMB-treatment. Notably, the cone arrestin *(Arr3)* gene was strongly downregulated in the cones of vehicle-treated rd10 mice, but not in the TMB group (Fig. 4F).

### Monotherapies with tamsulosin, metoprolol, or bromocriptine lack drug efficacy in rd10 mice

Earlier data based on a light-induced RD paradigm indicated a synergistic therapeutic effect when T, M, and B were used in combination[19]. It was therefore important to test T, M and B monotherapies, as well as TM, TB, and MB dual therapies, in the models of progressive and chronic RD. These experiments focused on a single time-point (P36) and utilized photopic ERG amplitude, ONL thickness, and retinal-detachment assessments as endpoints. The same study design for rd10 mice was utilized, as described above (Fig. 1A). None of the T, M, or B drugs alone exerted any signs of therapeutic effect in this robust paradigm (Fig. 5A–L). Monotherapy by M in fact exacerbated retinal detachment, indicating increased subretinal edema (Fig. 5H). Dual-therapy with T and B lacked therapeutic effects (Fig. 5M–P). In contrast, both dual therapies that included M (MB and TM) were effective in improving photopic ERG amplitudes (Fig. 5Q–X). The combination of adrenergic blockers (TM treatment) protected against M-cone- and S(UV)-cone-mediated retinal dysfunction effectively (Fig. 5Q, R). When

M was combined with B, an ergot alkaloid dopamine receptor-2 agonist, cone-pathway dysfunction (Fig. 5U, V) and also thinning of the ONL were significantly mitigated (Fig. 5W). Treatment with the tri-part TMB combination was more effective than the dual MB therapy against cone ERG decline (Fig. 5U, V).

### TMB slows cone-photoreceptor decline in the Rho^P23H/WT mouse model of autosomal-dominant RP

As the focus of this research is to establish an etiology/mutation-agnostic DMT for RD, we needed to test drug efficacy in an RP model with a clearly distinct pathological mechanism. For this purpose, we selected *Rho*^P23H/WT heterozygous mice that carry a missense mutation in the rhodopsin gene. TMB treatment of cyclic light-reared *Rho*^P23H/WT female mice was started at P21, and drug efficacy was tested in 7-month-long trials (Fig. 6A). At relatively early disease stages (1.5-3-month-old), the macrostructure of the retina in *Rho*^P23H/WT mice, assessed by OCT imaging, appeared similar between the treatment groups (Fig. 6B, C). However, scotopic ERG a-wave amplitude, arising from rod photoreceptor activation[44], was improved by 60% in the TMB group compared to the vehicle group at 3 months of age (Fig. 6D, E). No significant difference was observed for the b-wave (Fig. 6F). ANOVA analysis revealed a significant treatment effect on the ONL-thickness parameter when the entire 7-month duration of treatment was considered in a treatment cohort that was repeatedly measured throughout the experiment (Fig. 6G). ONL was on average 20% thicker in the TMB group at 8 months of age. Figure 6H shows photopic ERG waveforms of one *Rho*^P23H/WT mouse cohort, and healthy WT mice for comparison. The M-cone- and S(UV)-cone-mediated ERG response amplitudes did not differ significantly for vehicle- and TMB-treated *Rho*^P23H/WT mice at 3 or 5 months of age but did differ significantly at 6 and 8 months of age (Fig. 6I, J). The mean amplitude difference

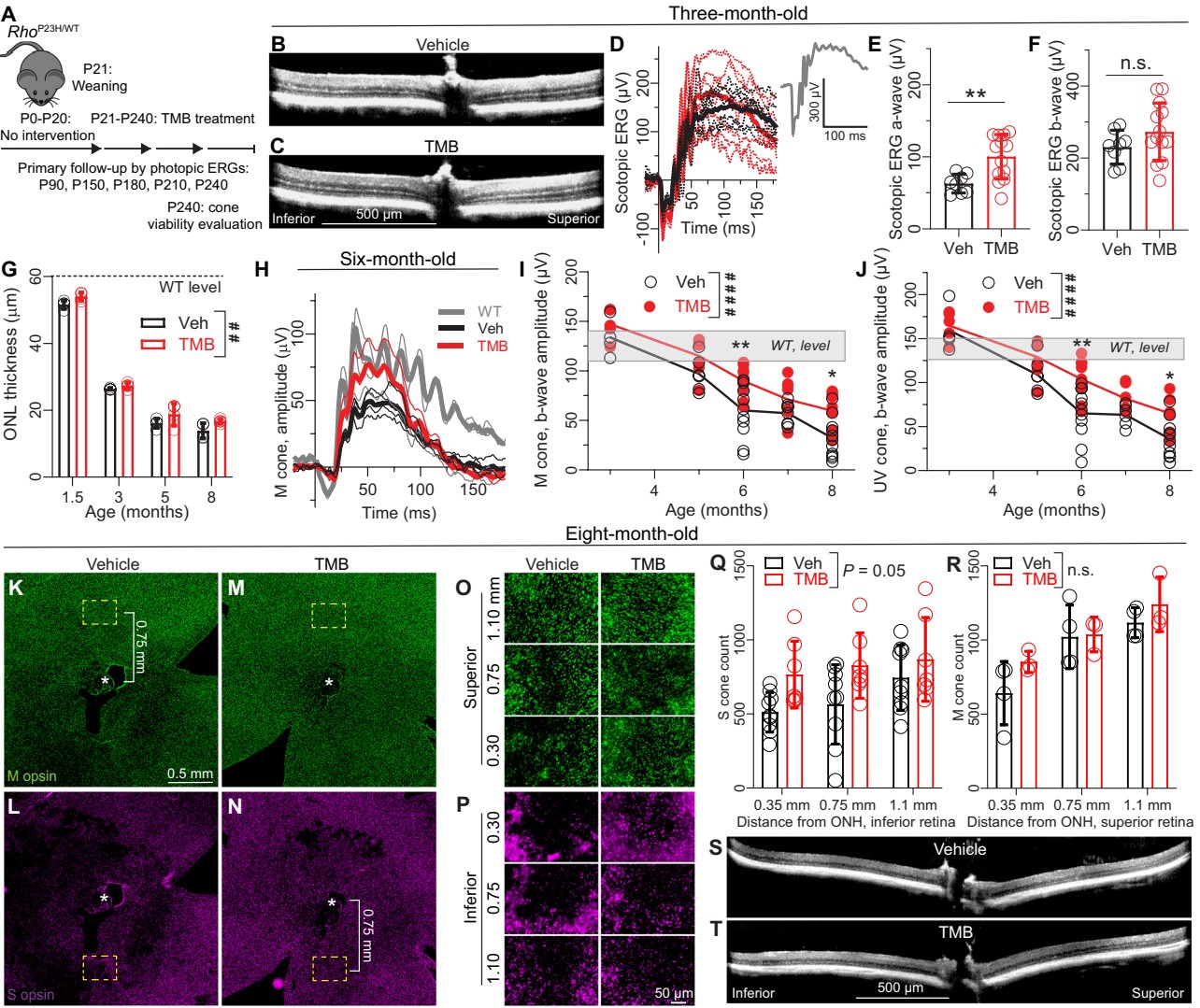

**Fig. 6 | Dietary TMB administration slows retinal degeneration in female $Rho^{P23H/WT}$ mice. A** Study design. **B–C** Representative OCT images at 3-months of age. **D** Representative group-averaged (thick lines) scotopic ERG waveforms (log cd·s/m² = 1.7). Thin traces represent individual mouse responses. **E–F** Scotopic ERG a-wave **E** and b-wave **F** amplitudes. **G** Longitudinal ONL thickness follow-up throughout the study in one treatment cohort (repeated measures study design). Data analyzed from OCT images. **H** Representative group-averaged (thick lines) photopic ERG waveforms at 6-months of age. Stimulus was elicited with a green LED that stimulates primarily the M cones (log cd·s/m² = 2.5). **I** M-cone **J** and S(UV)-cone-dominant photopic ERG b-wave-amplitudes. A repeated cross-sectional study. **K, L** Optic nerve head (ONH)-centered (* shows ONH) central retina images showing M-cone **K** and S(UV)-cone **L** populations in a vehicle-treated retina. Dashed-line yellow rectangles indicate 0.75 mm sampling site location. **M, N** ONH-centered central retina images showing M-cone **M** and S(UV)-cone **N** populations in a TMB-treated retina. **O–P** Representative M-cone **O** and P UV/S-cone images in superior and inferior retina, respectively, at different distances from the ONH. **Q–R** Inferior retina S(UV)-cone count analysis **Q** and superior retina M-cone count analysis **R**. Counting windows sizes were: width 360 μm, height 270 μm. **S, T** Representative OCT images. Statistical analysis performed for data in graphs E, and F was by Welch´s t-test (two-tailed). Data in graphs **G, Q** and **R** were analyzed by two-way RM ANOVA with the Geisser-Greenhouse correction. Data in graphs **I** and **J** were analyzed by ordinary two-way ANOVA. ANOVAs were followed by Bonferroni posthoc tests. T-test and Bonferroni post hoc results: *$P < 0.05$, **$P < 0.01$. ANOVA between-subjects main effects: ##$P < 0.01$, ####$P < 0.0001$. Data are presented as mean ± SD. Source data are provided as a Source Data file.

between vehicle- and TMB-treated $Rho^{P23H/WT}$ mice at 3 months of age was 10% and 3%, and at 8 months of age 86% and 80%, in response to the green and UV stimulation, respectively.

When the in vivo study was terminated at 8 months of age, retinal flat mounts were prepared and IHC was performed to assess M- and S(UV)-cone populations (Fig. 6K–N). M-cone populations were sampled in the superior retina, and S(UV)-cone populations in the inferior retina, similarly as with the dark-reared rd10 mice (Supplementary Fig. 6). Representative data for both groups of mice are shown in Fig. 6O and 6P–S(UV)-cone degeneration was apparent in both groups, but the S(UV) counts close to the central retina (0.3–0.75 mm from ONH) were around 50% higher in the TMB group compared to the vehicle group (Fig. 6Q). M-cone populations appeared better

preserved in both groups (Fig. 6K, M, O, R). Our sampling method did not find any treatment effect against M-cone degeneration (Fig. 6R). By the time of study termination, the retinas and particularly the ONL layers of the $Rho^{P23H/WT}$ mice were very thin (Fig. 6S, T), suggesting practically complete rod degeneration.

**TMB increases rod bipolar-cell light responses, improves behavioral-contrast sensitivity, and attenuates the alterations in the retinal transcriptome of the $Rpe65^{-/-}$ mouse model of LCA**
In addition to the mouse models of RP, we tested TMB treatment with the $Rpe65^{-/-}$ mouse model of Leber Congenital Amaurosis (LCA) (Fig. 7A). The phenotype in this LCA model is distinct from that of archetypical RP; in $Rpe65^{-/-}$ mice the cone photoreceptors are primarily

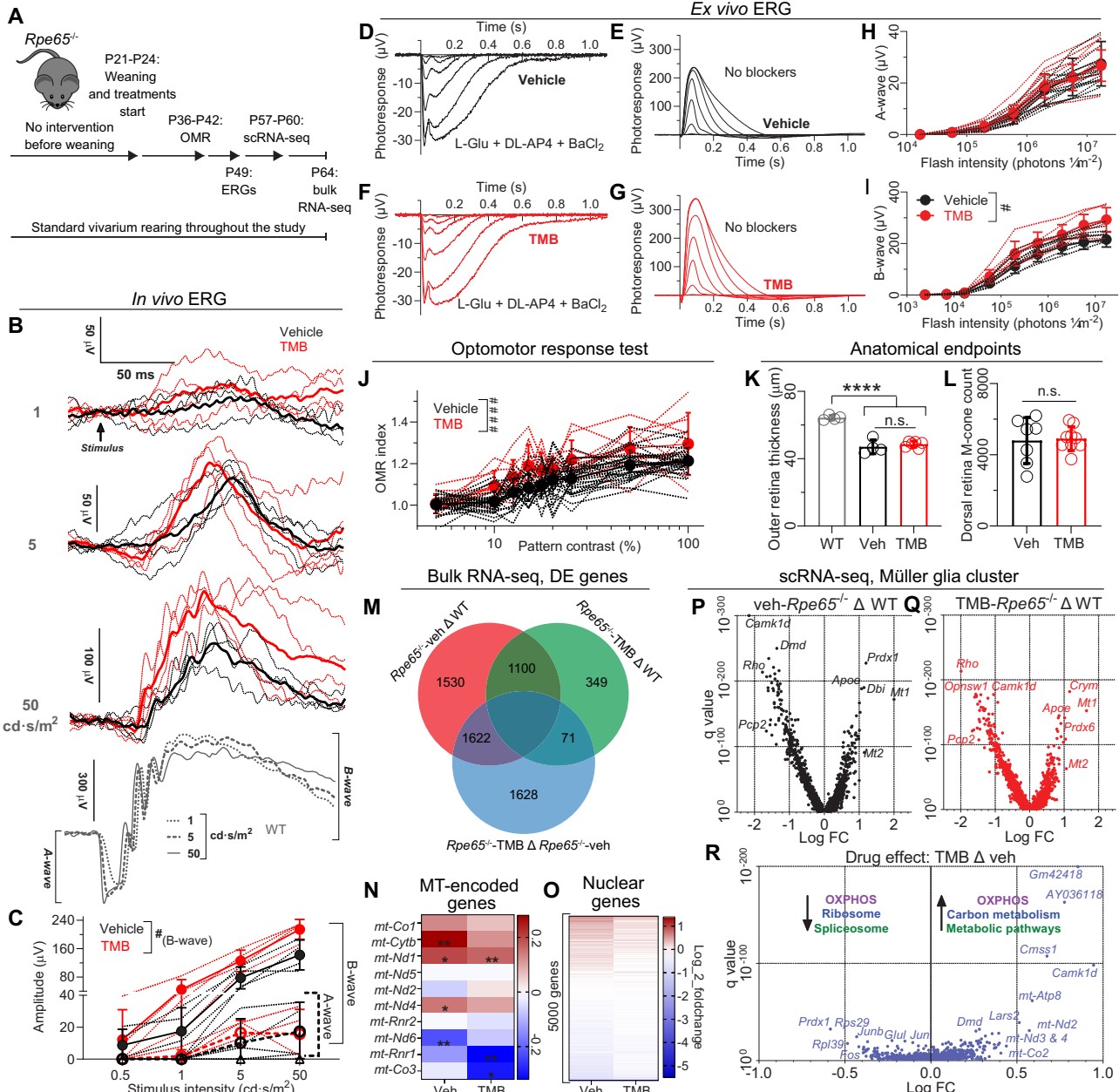

**Fig. 7 | TMB administration stabilizes retinal transcriptome and improves visual function in Rpe65⁻/⁻ mice. A** Study design. **B** Group-averaged scotopic ERGs. Thin lines represent responses from individual mice. **C** Scotopic ERG a- and b-wave amplitudes. **D**–**I** Ex vivo ERG. The perfusion medium was supplemented with synaptic blockers (vehicle, $n = 6$; TMB, $n = 7$ mice), or not (vehicle, $n = 6$; TMB, $n = 6$ mice). **H**–**I** Averaged intensity-response functions for ERG a-wave **H** or b-wave **I** recorded in the presence or absence of synaptic blockers, respectively. **J** Optomotor responses at different pattern contrasts. **K** Outer retina thickness as measured from OCT images. **L** Whole dorsal retina M-cone counts from retinal whole mounts. **M** Venn diagram of differentially expressed genes in retinal bulk RNA-seq. Comparison shown between WT, *Rpe65⁻/⁻*-vehicle, and *Rpe65⁻/⁻*-TMB groups. **N** Expression heatmap of 10 most highly expressing mitochondrial-encoded genes. The asterisks in heatmaps indicate significant difference (adjusted for multiple comparisons) compared to expression in WT mice. **O** Expression heatmap of 5000 most highly expressing nuclear-encoded genes. For clarity of presentation outlier genes *Gm473S* and *Eno1b* were removed. **P**, **Q** Volcano plots of DE genes between WT and vehicle-treated *Rpe65⁻/⁻* mouse MG cells **P**, or WT and TMB-treated *Rpe65⁻/⁻* mouse MG cells **Q**, from scRNA-seq data. Each *Rpe65⁻/⁻* group data was derived from cell suspensions combining seven retinas, whereas the WT group consisted of three retinas. Analysis was performed from 4,135 cells of the WT mice; 3979 cells from vehicle-treated; and 5,901 cells from TMB-treated *Rpe65⁻/⁻* mice. **R** DE gene comparison of *Rpe65⁻/⁻*-vehicle and *Rpe65⁻/⁻*-TMB groups in MG cells. Selected enriched KEGG pathways are shown. Data in graphs **C**, **H**, **I** and **J** were analyzed using RM two-way ANOVA with the Geisser-Greenhouse correction, whereas data in graph **K** were analyzed by 1-way ANOVA followed by the Bonferroni post hoc test: ****$P < 0.0001$. Data in graph **L** were analyzed by Welch´s t-test. The pound signs indicate an ANOVA-significant main effect between treatments: #$P < 0.05$, ####$P < 0.0001$. Data are presented as mean ± SD. Source data are provided as a Source Data file.

affected (Supplementary Fig. 8) - they die quickly and do not display light responses[45]. In contrast, rod degeneration in *Rpe65⁻/⁻* mice is slow, although the responses of visual chromophore-deficient rods to light are severely attenuated.

Scotopic ERG recordings after 1-month on treatment revealed more robust light responses with the TMB-treated than with the vehicle-treated cyclic light-reared *Rpe65⁻/⁻* mice (Fig. 7B, C). The mean ( ± SD) b-wave amplitude in the vehicle-group was 142 ± 42 μV and in

TMB-group $214 \pm 28$ μV in response to the highest intensity flash (50 cd·s/m²). It should be noted that the ERG responses of $Rpe65^{-/-}$ mice are markedly diminished compared to the responses of WT mice, and the a-wave in the in vivo ERG recording of $Rpe65^{-/-}$ mice is practically indistinguishable from the background signal (Fig. 7B). To dissect whether TMB increases the responses of the photoreceptors or ON bipolar cells in $Rpe65^{-/-}$ mice, we turned to ex vivo ERG recording from isolated retinas (Fig. 7D–I)[46]. We found that while a-wave responses were identical in vehicle- and TMB-treated $Rpe65^{-/-}$ mice (Fig. 7D, F, H), depolarizing b-wave responses were improved by TMB (Fig. 7E, G, I). The mean maximum b-wave amplitude in the vehicle-group was $215 \pm 31$ μV and in the TMB-group $293 \pm 51$ μV, suggesting that TMB´s therapeutic effect is occurring at the inner retina level in $Rpe65^{-/-}$ mice. To study if this improvement in the ERG b-wave translates into enhanced visual behavior, we used the OMR test. Indeed, TMB-treated $Rpe65^{-/-}$ mice responded better to drifting vertical gratings of various contrast than did vehicle-treated $Rpe65^{-/-}$ mice (Fig. 7J). At 100% contrast, the vehicle-group´s mean ( $\pm$ SD) OMR index was $1.21 \pm 0.07$, and TMB-group´s mean was $1.30 \pm 0.15$. Lack of significant differences in the analyzes of ONL thickness (Fig. 7K; Supplementary Fig. 8A–C) or preservation of the M-cone population (Fig. 7L; Supplementary Fig. 8D, E) further suggested the absence of TMB effect at the level of photoreceptors in $Rpe65^{-/-}$ mice.

To gain insight regarding the mechanism of the selective TMB-treatment effect at the inner retina level in $Rpe65^{-/-}$ mice, we treated a subset of the mice for 6-weeks and collected retinas for bulk RNA-seq analysis at P64. Several findings indicated that the retinal transcriptome in TMB-treated $Rpe65^{-/-}$ mice was stabilized closer to that of WT (Fig. 7M–O; Supplementary Data 10), like what was observed for RNA-seq analysis with the rd10 mice (Fig. 1I–L). First, we found many fewer DE genes in the comparison of the TMB-$Rpe65^{-/-}$ group *versus* WT (349 genes) than for the vehicle-$Rpe65^{-/-}$ group *versus* WT (1530 genes) (Fig. 7M; components of the Venn diagram are shown in Supplementary Data 11). Considering mitochondrial genes, 5 out of 10 were significantly altered in vehicle-treated $Rpe65^{-/-}$ mice *versus* WT, whereas 3 out of 10 were significantly changed in the TMB-treated $Rpe65^{-/-}$ mice (Fig. 7N; Supplementary Data 12). Analogous to the Venn diagrams (Fig. 7M), the heatmaps suggested that TMB treatment led to adjustment of the transcriptome in $Rpe65^{-/-}$ mice back toward WT levels (Fig. 7O). For example, the expression of catecholaminergic signaling-associated genes was mostly maintained closer to the WT level in the TMB-treated $Rpe65^{-/-}$ mice *versus* vehicle-treated animals (Supplementary Fig. 9A). We also performed gene-set enrichment analysis (GSEA) to test which signaling pathways may be modulated by TMB. The GSEA analysis identified the relevant categories of "visual perception" and "mitochondrial protein-containing complex" as being altered in the direct comparison of vehicle-$Rpe65^{-/-}$ group *versus* the TMB-$Rpe65^{-/-}$ group (Supplementary Fig. 9). Thus, several genes associated with visual perception were excessively dysregulated in vehicle-treated $Rpe65^{-/-}$ mice but to a lesser extent in TMB-treated $Rpe65^{-/-}$ mice (Supplementary Fig. 9B). The TMB-treatment effect was particularly clear in the mitochondrial function-associated gene set, wherein tens of genes encoding respiratory complex proteins showed dysregulation in the vehicle-treated $Rpe65^{-/-}$ mice but remained close to WT level in the TMB-treated $Rpe65^{-/-}$ mice (Supplementary Fig. 9C).

To distinguish which retinal cell classes were most affected in terms of transcriptomic changes, we performed an additional 5-week-long trial. Vehicle- or TMB-treatments were started in $Rpe65^{-/-}$ mice at P21-P24, and retinal samples were collected for scRNA-seq at P57-P60. Retinas from nontreated and age-matched WT mice served as healthy controls. The largest cell clusters identified through scRNA-seq were rods, RBCs, and MGs, in that order (Supplementary Figs. 10, 11). As also in rd10 mice, the MG cluster displayed the most extensive gene regulation in the $Rpe65^{-/-}$ mice (Supplementary Fig. 10). By amplitude,

gene regulation in the MGs appeared slightly more robust in the $Rpe65^{-/-}$-vehicle *versus* WT comparison (Fig. 7P), than in the $Rpe65^{-/-}$-TMB *versus* WT comparison (Fig. 7Q; supplementary Data 13, 14). Direct comparison of the MG-transcriptomes of the $Rpe65^{-/-}$-vehicle *versus* $Rpe65^{-/-}$-TMB revealed TMB-dependent downregulation of many cAMP-responsive genes, such as *Junb*, *Fos*, and *Jun* (third, fifth and eighth most downregulated genes, respectively; Fig. 7R, Supplementary Data 15). On the other hand, TMB treatment led to upregulation of several mitochondrially encoded genes and metabolism-related KEGG pathways in the MGs from $Rpe65^{-/-}$ mice (Fig. 7R). Interestingly, *Camk1d*, which was the most downregulated gene in the MGs with respect to the $Rpe65^{-/-}$-vehicle *versus* WT comparison (Fig. 7P), also showed the highest upregulation and fold-change after TMB treatment in $Rpe65^{-/-}$ mice (Fig. 7R). Comparison of the transcriptomes of rods and RBCs also indicated a trend towards metabolic improvement in $Rpe65^{-/-}$ mice after TMB treatment (Supplementary Fig. 12A, B), which could explain the improved rod-pathway function in these mice (Supplementary Fig. 12C–J).

## Monotherapies with tamsulosin, metoprolol, or bromocriptine lack efficacy against retinal dysfunction in Rpe65$^{-/-}$ mice

To assess whether mono- or dual-therapies using T, M and/or B could also lead to improved retinal function in $Rpe65^{-/-}$ mice, we performed OMR behavioral tests and scotopic ERG recordings after 3-4 weeks on the treatments. Analogous to the results from the rd10-mouse experiments (Fig. 5A–L), monotherapies of the $Rpe65^{-/-}$ mice with T, M or B did not show any efficacy against retinal dysfunction (Fig. 8A–F). TB dual-therapy was also not effective for the rd10 mice (5M-P), but showed a significant, albeit small, OMR behavioral improvement with the $Rpe65^{-/-}$ mice (Fig. 8G). TM treatment clearly led to enhanced OMR behavior (Fig. 8I), but ERG amplitudes were not improved (Fig. 8J). MB treatment led to improvement in both OMR behavior (Fig. 8K;) and ERG amplitudes (Fig. 8L). As expected, combination of the three drugs, TMB, increased both OMR behavior (Fig. 8M and ERG amplitudes (Fig. 8N). Importantly, TMB led to higher OMR and ERG responses close to sensitivity threshold. At 10% pattern contrast (Fig. 8M), the OMR index of the vehicle-group was $1.017 \pm 0.052$ and for the TMB-group it was $1.093 \pm 0.072$. At 0.5 log cd·s/m² ERG stimulus intensity (Fig. 8N,), the vehicle-group´s b-wave amplitude was $54 \pm 34$ μV and for the TMB-group it was $77 \pm 27$ μV.

## Long-term subcutaneous infusion of TMB improves cone viability in the PDE6A$^{-/-}$ dog model of autosomal-recessive RP

We also applied our long-term TMB treatment approach to a large animal model of RP, the $PDE6A^{-/-}$ dogs. Like most mammalian species, dogs do not have macula with a central fovea but do have an equivalent cone-rich area, called the area centralis, centered around 3.8 mm dorso-temporal from the ONH[47].

Pilot experiments showed dietary TMB administration to be impractical for dogs, so we used subcutaneous infusion with implantable mini pumps. In one set of experiments, implants were inserted at 1-month of age and replaced every 2 weeks; and the entire experiment lasted ~7 months (Fig. 9A). Figure 9B shows quantification of drug levels in serum from 3 dogs during the first 7 weeks of the trial; and Fig. 9C shows data for a representative dog throughout the duration of the study. Mean levels of T in serum samples from dogs ($3.7 \pm 1.8$ ng/ml, first two weeks of experiment) were found to be ~3-fold lower than T levels in mouse serum (Fig. 1B). In mouse serum, levels of B were below reliable detection limit (1 ng/ml), whereas with dogs the detection limit was more often exceeded (first two week´s average, $1.5 \pm 0.7$ ng/ml; Fig. 9C). A particularly large difference in serum drug levels between the mouse and dog experiments was observed for M. In mice, average levels of M ranged from $759 \pm 644$ to $2361 \pm 614$ ng/ml, whereas in dogs the average level was $17.0 \pm 8.4$ ng/ml during the first two weeks of the experiment. In humans, M levels are typically in the

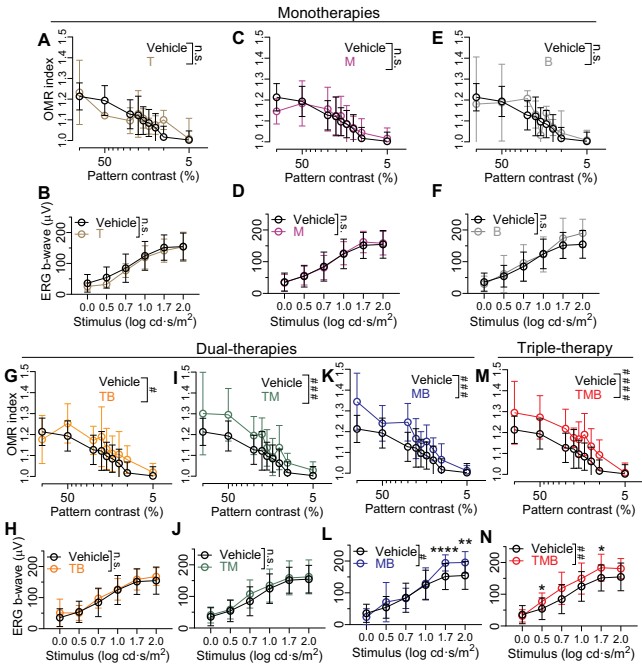

**Fig. 8 | Tamsulosin, metoprolol, or bromocriptine monotherapies are not effective in improving retinal function in Rpe65⁻/⁻ mice.** Treatments were started at P21, and treatment effect was tested after 3–4 weeks of drug therapy. The drug effect was evaluated using two different methods: Scotopic ERG b-wave and optomotor response. The same vehicle-group data are used for comparison with each of the data sets for all treatments. Data in M is duplicated from Fig. 6J. **A**, **B** Tamsulosin (T) monotherapy data. **C**, **D** Metoprolol (M) monotherapy data. **E**, **F** Bromocriptine (B) monotherapy data. **G**, **H** Tamsulosin and bromocriptine (TB) dual-treatment data. **I**, **J** Tamsulosin and metoprolol (TM) dual-treatment data. **K**, **L** Metoprolol and bromocriptine (MB) dual-treatment data. **M**, **N** Tamsulosin, metoprolol, and bromocriptine (TMB) triple-treatment data. Statistical analysis for all data was performed using repeated measures two-ANOVA with the Geisser-Greenhouse correction followed by Bonferroni posthoc tests: *$P < 0.05$, **$P < 0.01$, ****$P < 0.0001$. Pound signs denote ANOVA between subject main effect significance levels: ##$P < 0.01$, ###$P < 0.001$, ####$P < 0.0001$. Data are presented as mean ± SD. Source data are provided as a Source Data file.

range of hundreds rather than tens of ng/ml during treatment of cardiovascular conditions[37].

Overall, we reached reasonable T and B levels via subcutaneous infusion in the dogs, but the M component was severely underdosed. To mitigate underdosing, we altered the study protocol halfway through the project and doubled the infusion by inserting two implants per surgery from the fifth surgery onwards (after 2 months on treatment). We also refined the protocol by inclusion of vehicle administration via pumps in control dogs. However, neither of these refinements notably altered any of the important endpoints measured in the study (Supplementary Fig. 13). Accordingly, results from all cohorts were pooled for the rest of the analyzes to improve statistical power.

Photopic ERGs were recorded monthly to follow cone-pathway function throughout the experiment. The ERG responses of *PDE6A⁻/⁻* dogs were prominently delayed compared to those of unaffected heterozygous *PDE6A⁺/⁻* dogs (Fig. 9D, E). ERG b-wave amplitudes were initially larger in *PDE6A⁻/⁻* dogs but progressively declined towards the end of the study (Fig. 9D–F and Supplementary Fig. 14). Slightly hastened b-wave kinetics (*i.e.*, decreased b-wave peak time) for the TMB-treated *PDE6A⁻/⁻* dogs provided the first evidence of drug efficacy (Fig. 9E; control mean ± SD = 38.00 ± 1.05 ms, TMB mean ± SD = 36.13 ± 0.82 ms). However, b-wave amplitudes did not significantly differ between TMB-treated and control *PDE6A⁻/⁻* dogs (Fig. 9F). At the end of the trials, retinal flat mounts were prepared and stained with a cone

marker, peanut agglutinin (PNA). At this disease stage, cone population in the peripheral retina was severely degenerated, particularly in the inferior part of the retina (Fig. 9G–I & Supplementary Fig. 15). We sampled cone populations in four inferior retina locations and in seven superior retina locations (Supplementary Fig. 15). The area centralis was also imaged. Supplementary Figs. 16 and 17 show inferior and superior middle retina as well as area centralis images from individual dogs. Semi-automated counting revealed a higher PNA-punctate count for TMB-treated *PDE6A⁻/⁻* dogs than for control *PDE6A⁻/⁻* dogs in the superior retina towards the periphery (Fig. 9J; Supplementary Fig. 16). At 7.5 mm distance from the ONH, the TMB-group's mean PNA puncta count for the TMB-group (439 ± 93) was 79% higher than that for to the vehicle -group (245 ± 141). In contrast, in the inferior retina or area centralis, the PNA-punctate count did not differ between the two groups (Fig. 9I, J, K, L; Supplementary Figs. 16-17).

## Discussion

In the current study, we utilized a systems pharmacology-based approach and targeted retina-expressing $G_q$-, $G_s$- and $G_i$-coupled GPCRs[19] simultaneously, using clinically approved drugs tamsulosin (T), metoprolol (M), and bromocriptine (B) as prototype compounds. Also other alpha- and beta-receptor blockers[19,26,48] and D2-like receptor agonists[49] have shown protective properties against RD, and could potentially be used in the context. Our goal was to attain a sustained retinal-protective effect in vivo in several etiologically distinct chronic RD models via dietary administration (in mice) or subcutaneous infusion (in dogs). Despite the heterogeneity of the disease models, and the inability to control dosing precisely, TMB combination treatment was effective in all the tested paradigms. The *Pde6β*rd10 mutation in rd10 mice leads to instability and dysfunction of PDE6, increased free cGMP, subsequent opening of cGMP-gated channels, and increased $Ca^{2+}$ influx into rods, which is detrimental for them[50]. The *Rho*P23H/WT mutation is characterized by early rhodopsin misfolding and subsequent ER stress[51]. A common feature of the RP phenotype is primary rod photoreceptor death, and cones die much later[8–11]. This was evident in our experiments as RP-affected animals retained moderate cone populations until advanced disease states. For instance, we observed relatively high cone densities in the superior retina and discernible area centralis in 8-month-old *PDE6A⁻/⁻* dogs despite their rods having mostly degenerated by 5 months of age[52].

With DMTs that do not directly address the cause of rod degeneration, preventing collateral damage of the cones is more feasible, as evidenced by our data. Nevertheless, the positive effects of TMB treatment were not restricted to the cone system. Rod-dominant scotopic ERGs were improved by TMB treatment in dark-reared rd10 mice, and in young-adult *Rho*P23H/WT mice, suggesting that the surviving rods sustained better vitality under TMB treatment, even if the treatment effect was minor-to-nonexistent with respect to ONL thinning (an anatomical marker of rod degeneration). More straightforward evidence that TMB treatment can also improve rod-pathway function was obtained from experiments with the *Rpe65⁻/⁻* mouse model of LCA. In *Rpe65⁻/⁻* mice, the cones completely lack functionality[45] and die quickly after opening of the eyes, due to cone-opsin mislocalization in the absence of 11-*cis*-retinal production and supply by the RPE[53]. In contrast, rods die slowly in *Rpe65⁻/⁻* mice as their opsin is more stable[54], and they retain a residual light responsivity despite severe chromophore insufficiency. We observed improved scotopic ERG b-wave amplitudes in TMB-treated *Rpe65⁻/⁻* mice, but we were unable to reliably detect a-waves from the background signal. Therefore we also recorded ERGs in ex vivo conditions wherein synaptic transmission from rods to bipolar cells can be reliably blocked pharmacologically yielding pure and selective photoreceptor responses in the isolated retina[46]. The ex vivo ERG revealed a TMB-dependent functional improvement in the *Rpe65⁻/⁻* mouse rod ON bipolar cells rather than the photoreceptors.

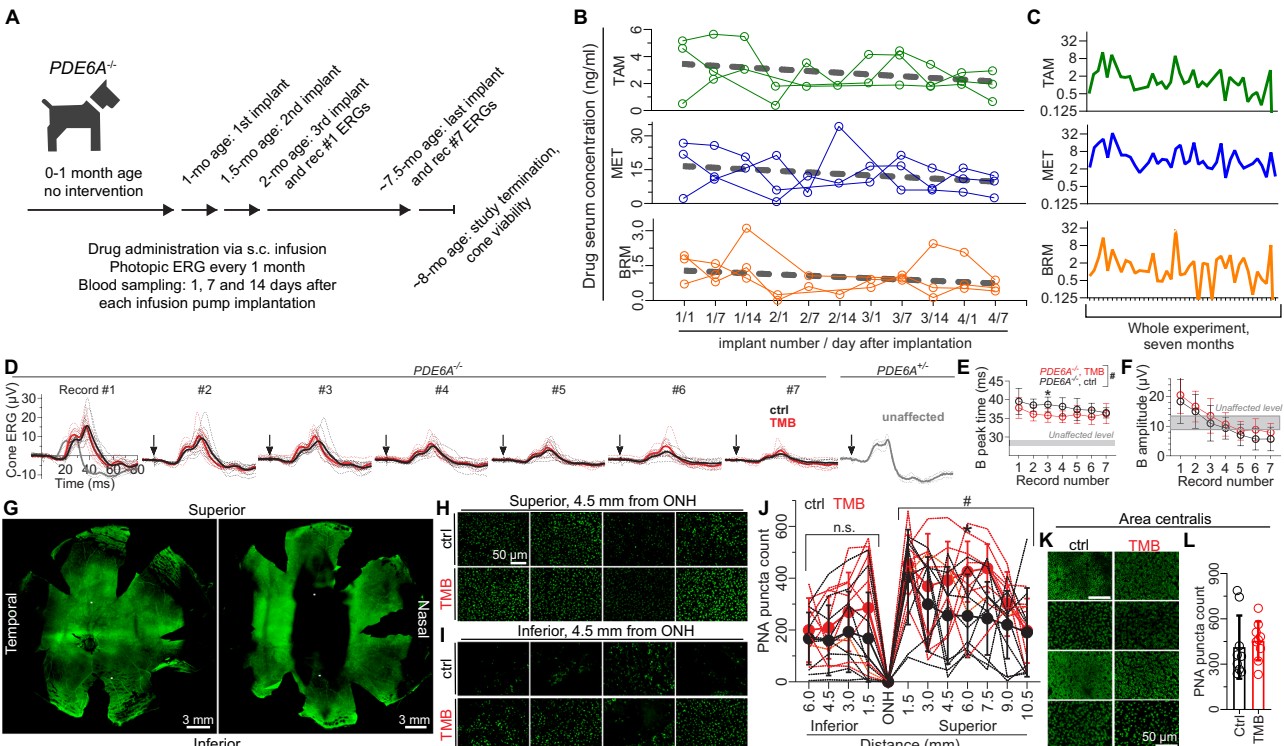

**Fig. 9 | Subcutaneous TMB infusion improves cone viability in PDE6A$^{-/-}$ dogs.**
**A** Study design. **B** Drug serum levels during the first 7 weeks of study in three dogs. Dots and lines represent data from individual dogs. Gray dashed line shows linear regression. Outliers were removed using the ROUT method with FDR set at 0.5%. **C** Drug serum level follow-up throughout the study in subject #3. **D** Group-averaged photopic ERG waveforms of each monthly recording. The ERGs are contrasted to PDE6A$^{+/-}$-carrier dogs without RP phenotype. Black, vehicle $n = 10$; red, TMB $n = 9$; gray, unaffected ctrl (PDE6A$^{+/-}$, $n = 3$). **E, F** Photopic ERG b-wave peak time **E** and amplitude **F** follow-up. **G** Representative dog retina whole mounts stained with a cone-marker peanut agglutinin (PNA). **H, I** PNA puncta counting windows (width 240 μm, height 180 μm) in the superior middle retina **H**, upper row controls and lower row TMB) and in **I** the inferior middle retina in litters 6-8 (note, these cohorts were vehicle-controlled). **J** PNA puncta count. Semi-automated counting was performed by a blinded observer. All litters 1-8 were included in the analysis. Due to distinctly differential degeneration in inferior *versus* superior retina, PNA puncta count in inferior and superior retinal sides were analyzed separately. Vehicle, $n = 10$; TMB, $n = 9$. **K** Representative area centralis (AC) images that were used as PNA puncta counting windows (150 μm, height 110 μm). Images from dogs in litters 6–8. **L** PNA puncta count in AC. Counting in this region was performed manually by a blinded observer. Litters 1–8 were included in the Mann-Whitney U-test (two-tailed) analysis (ctrl, $n = 9$; TMB, $n = 9$; n.s.). Statistical analysis in **E, F** and **J** was performed by two-way repeated measures (**E, F, J** at superior retina) or mixed-model (**J** at inferior retina) ANOVA with Geisser-Greenhouse correction and followed by Bonferroni post hoc tests. The pound signs signify a significant between-subjects main effect: $^{\#}P < 0.05$. The asterisk * signify $P < 0.05$ in Bonferroni test. Data are presented as mean ± SD. Source data are provided as a Source Data file.

The ex vivo recording also helped to rule out the possibility that TMB's positive effects arise from direct vascular modulation. Importantly, behavioral visual function was also improved in TMB-treated Rpe65$^{-/-}$ mice, despite the lack of treatment effect on ERG a-waves. Taken together, the outcomes from several distinct IRD paradigms indicate that TMB's therapeutic effect arises from complex targeting of various retinal cell types, including those at the inner retina level.

Mechanistic investigation of therapeutic effects preferentially should be conducted in an early state of degenerative disease so that causative rather than consequent effects of drug treatment could be more reliably detected. We did not yet detect structural differences in the retina between the dark-reared vehicle- and TMB-treated rd10 mice when they were 2-months of age, but scotopic ERG indicated a modest positive treatment effect on function. We decided to perform investigations on retinal oxidative stress in analogous settings. Due to high polyunsaturated fatty acid (PUFAs) content, the retina is particularly vulnerable to lipid peroxidation that can occur before the onset of significant photoreceptor degeneration[55]. We found ~2-fold increase in 4-hydroxynonenal (4-HNE) content, a standard lipid peroxidation biomarker derived from n-6 PUFA oxidation, in vehicle-treated rd10 mice compared to WT. This effect was blocked by the TMB regimen suggesting that mitigation of oxidative stress contributes to its therapeutic effect. 4-HNE can trigger an adaptive increase in the expression

of catalase which is a major hydrogen peroxide-decomposing enzyme[56]. Indeed, catalase was increased in concert with the 4-HNE content in rd10 mouse retinas, but this effect was also ablated by TMB treatment.

The same dark-reared rd10 paradigm served well for investigation of transcriptomic regulation which we assayed at the single cell level in this case. ScRNA-seq was performed also in Rpe65$^{-/-}$ mice after 5-week-long TMB treatment. Due to rod death in rd10 mice, the largest cell cluster identified in these mice was the Müller glial cells, the main glia cells in the retina which have a crucial role in maintaining metabolic support, homeostasis, and tissue integrity. MGs are known to respond robustly to retinal stress, and this response can be either protective or detrimental to retinal function. Indeed, MGs showed massive differences in transcriptomic regulation between mutant and WT mice in our analyzes. TMB significantly affected the expression of almost 500 genes in the MGs of rd10 mice, and almost 1000 genes in the MGs of Rpe65$^{-/-}$ mice. In the rd10 model, marked interconnectedness of the downregulated genes was observed, as most of the genes were connected through a single network, which was largely associated with decreased cellular response to stress. In contrast, TMB-mediated upregulated genes in the MGs of rd10 mice were organized into five distinct functional networks that were associated with oxidative phosphorylation, ribosome, core histone, circadian rhythm, and

phototransduction. In *Rpe65*-/- mice, TMB treatment enhanced mitochondrial gene expression and metabolism-associated gene sets in rod, RBC and MG clusters, which may explain the improved visual function as detected by ERGs and OMR.

In rods from rd10 mice, surprisingly few differences compared to WT were detected, even though *Pde6b* is expressed exclusively in these cells. A larger number of DE genes was observed in the RBCs. Notably less dysregulated genes were observed in the TMB-treated than in the vehicle-treated rd10 mouse RBCs, suggesting that TMB treatment helped to maintain RBC homeostasis.

In addition to MGs, the direct comparison of vehicle- and TMB-treated rd10 mouse groups demonstrated relevant TMB-mediated transcriptomic regulation at the level of the ON-CBCs and the cones. In the largest ON-CBC cluster 3, many genes involved in oxidative phosphorylation showed downregulation in the vehicle-treated group compared to the TMB-treated group. Three genes belonging to the respiratory chain complex IV, or cytochrome c oxidase, showed a similar direction of regulation in the cone cluster. Some major genes that are upregulated in response to cAMP were dramatically upregulated in rd10 mouse cones compared to WT. TMB treatment mitigated the overexpression of *Junb*, *Fos*, and *Fosb* in the cones of rd10 mice and in the MGs of *Rpe65*-/- mice, which is consistent with the primary pharmacologic actions of M and B. Another prominent gene that was differentially expressed in the vehicle- *versus* TMB-treated rd10 mouse cones was *Cartpt*, which is positively regulated by dopaminergic activity[57]. As *Drd4* also showed distinctly differential regulation, it is conceivable that dopaminergic activity in cones was modified by the TMB treatment. Notably, bulk retina RNA-seq performed on vivarium-reared rd10 mice and *Rpe65*-/- mice also demonstrated a dysregulated catecholaminergic system. Several genes encoding adrenergic and dopamine receptors as well as enzymes that degrade catecholamine neurotransmitters were upregulated in the rd10 mouse retina bulk RNA-seq. In *Rpe65*-/- mice, dysregulation in the corresponding genes was more bidirectional.

While the rod-system in mice closely resembles that of humans, their cone-system is markedly different. Mice are nocturnal animals, and instead of having an S-cone that would optimally respond to blue wavelengths of light, their "S-cones" are tuned to UV spectra[58]. The C57BL mouse S(UV)-cone and M-cone populations display a distinct density gradient, so that the S(UV)-cone opsins are expressed in the inferior side of the retina, with M-cone opsins in the superior side; but there is very little central-to-periphery gradation[59]. In comparison, dog retina´s cone-system is closer to that of humans. Like most mammalian species, dogs do not have fovea. However, they do have a macula-equivalent cone-rich area, called the area centralis[47], and possess S- and M-cones similar to humans[47,60].

To advance the translational relevance of our preclinical data, we also tested TMB treatment effect in *PDE6A*-/- dogs. Photopic ERGs revealed severely slowed cone-pathway kinetics in *PDE6A*-/- dogs as their b-wave peaks occurred ~10 msec later compared to unaffected *PDE6A*+/- dogs. The TMB treatment positively affected the ERG kinetics and rendered responses faster, but it did not significantly improve ERG amplitudes. At the end of the trials, we detected better preserved cone populations, particularly in the middle superior retina, using PNA-positive cell counting from retina flat mounts. The area centralis was generally distinguishable, but there was no difference in PNA-positive cell counts between control and TMB-treated *PDE6A*-/- dogs in that region. The lack of effect on the area centralis cone population may explain the lack of significant treatment effect on the ERG amplitudes. Furthermore, our analyzes of drug levels in the serum revealed underdosing of M (< 20 ng/ml) compared to our mouse studies (average range of means: 759-2361 ng/ml) and to published clinical PK data in cardiovascular indications[37]. Importantly, our experiments with TM, TB, and MB dual-therapies conducted in rd10 and *Rpe65*-/- mice indicated that M was a crucial component in the combination

treatment strategies. In both mouse models, T, M or B monotherapies lacked any efficacy based on the selected outcome measures. With the dual therapies only the TM and MB treatments were therapeutic in rd10 mice. Since M was most important for efficacy within our therapeutic strategy, we postulate that M underdosing is the best explanation why TMB´s therapeutic effect was less apparent in *PDE6A*-/- dogs than in rd10 mice, although the difference could also be related to pathophysiological differences between the models. Notably, the phenotype in *PDE6A*-/- dogs is similar to rd10 mice but more severe, as *PDE6A*-/- dogs do not express any rod-specific PDE6 and completely lack rod function[61], whereas rd10 mice initially demonstrate both[39].

RDs characteristically manifest at a restricted retinal locus or cell type in the earliest stages of the disease, rod death in RP being a prime example of this feature. However, it is unlikely that the physiology in the rest of the retina remains intact. Besides stressors such as oxidative stress and inflammation that undoubtedly diffuse to collateral cell populations, photoreceptor death also induces pathological neurotransmitter metabolism, signaling, and electrochemical overactivity[62,63]. Specifically, some recent findings indicate that components of the sympathetic nervous system may be overactive in the retina during the degenerative state. Cammalleri et al. showed increased norepinephrine levels in rd10 mice[64], which could arise from local production[65] in response to inflammatory stimuli[66]. In our bulk RNA-seq data, many genes related to the catecholaminergic system show disease progression-dependent dysregulation in rd10 mice; for example, *Maoa* and *Comt* show upregulation. Although TMB treatment does not significantly alleviate this part of the transcriptomic dysregulation, it does directly target the catecholaminergic signaling at the receptor level. Notably, our RNA-seq data show low to nonexistent expression of dopamine- or norepinephrine-reuptake transporters (*Slc6a2* or *Slc6a3*) in the retina, which points to a particularly crucial role of MAOs and COMT in ceasing catecholaminergic activity. COMT was readily detectable by immunoblotting, and showed upregulation in rd10 mice compared to WT. Published data show that either $\alpha_2$-agonist drugs that decrease presynaptic norepinephrine release[20,29], or drugs that block postsynaptic $\alpha_1$- and/or $\beta_{1/2}$-receptors[19,26,67], are protective for the stressed retinal neurons, suggesting that overactivity of the catecholaminergic system could play a detrimental role in RD progression. Increased catecholamine neurotransmitter levels and activity could be harmful in several ways, for instance, through formation of neurotoxic free radicals from excessive catecholamine neurotransmitter metabolism, or sensitization of neurons to excitotoxicity[68]. Increased norepinephrine activity also directly elevates intracellular $[Ca^{2+}]$ through agonism at the $\alpha_1$- and $\beta$-receptors. Agonists of the dopamine-2-receptor (DRD2) such as bromocriptine could stabilize dopaminergic activity via autoregulation, as DRD2s are localized pre-synaptically in the dopaminergic amacrine cells that are the sole dopamine-producing cells in the retina[69]. The mechanisms of how catecholaminergic GPCR drugs exert their neuroprotective effects have remained enigmatic for decades. The fact that components of the dopamine and norepinephrine systems often overlap and can act in parallel[70] may help to explain the therapeutic efficacy of our combined TM, MB, and particularly TMB treatment, even if any of the three components (T, M or B) alone were not effective against RD. The link between the sympathetic nervous system and RD may open a paradigm for continuing investigation of neuroprotective mechanisms by catecholaminergic GPCR drugs.

## Methods

### Disease models and study design
Parameters of study groups in each experiment is presented in Supplementary Data 1. Most experiments were conducted with a commonly used *Pde6β*rd10 mouse model (aka rd10 mice;

RRID:IMSR_JAX:004297) of autosomal recessive RP. These mice carry a naturally occurring point mutation in the phosphodiesterase 6b (*Pde6β*) gene leading to extensive loss of rod photoreceptor nuclei by post-natal day 24 (P24) if mice are reared in vivarium conditions[39]. Light restriction/dark rearing slows RD in rd10 mice remarkably. Since the retinal disease characteristics and progression are different in rd10 mice reared in vivarium or full darkness (dark-rearing, DR)[43], we chose to perform drug efficacy trials under both of these distinct conditions. Each litter was randomly divided at weaning into vehicle group fed with standard chow (Prolab IsoPro RMH3000, LabDiet, St. Louis, MO, USA), and a drug group fed with drug-supplemented diet starting at P28 (see description below, "*drug administration*"). Feeding was *ad libitum*. In trials using vivarium conditions, the mice were dark-reared until they were transferred to vivarium at P29 (~24 hours after starting the special diets). The treatment effects were tested at 1 week, 2 weeks, or 4 weeks of treatment (see "*drug efficacy testing* in vivo…," below). In trials utilizing full DR, the mice were monitored monthly using a brief scotopic electroretinogram (ERG) and optical coherence tomography (OCT) imaging protocol, and dark conditions were maintained continuously, as much as possible. The brief ERG protocol consisted of two green flashes at 10 cd·s/m$^2$ with an inter-stimulus interval of 25 sec, and the responses were averaged for a- and b-wave analyzes. The experimenter used a dim red-light headlamp when performing the OCT in a darkened laboratory. Terminal experiments in the DR paradigm were performed either at P58 (1-month on treatment) or P120 (3 months on treatment).

We used a heterozygous P23H rhodopsin-mutation knock-in mouse line (*Rho*^P23H/WT^ mice; RRID: IMSR_JAX:017628) as an additional RP mouse model to test drug efficacy. These mice represent a model for the most common form of autosomal dominant RP in North America[71]. The *Rho*^P23H/WT^ mice represent an intermediately progressing rod degeneration, but long-preserving cone population[72]. Mouse litters were divided similarly as described above, and drug treatment was initialized at weaning (P21). The *Rho*^P23H/WT^ mice were reared under vivarium conditions. Drug efficacy monitoring was mainly focused on the cone system, using photopic ERGs (see "*Drug efficacy testing* in vivo by ERG and OCT") that were performed first at 3 months of age, and monthly starting at 5 months of age until ~8 months of age. In a subset of experiments, OCT was performed at 1.5, 3, 5, and 8 months of age, and scotopic ERG at 3 months of age. Terminal experiments were performed at 8 months of age.

*Rpe65*^-/-^ mice (a kind gift from Dr. Michael Redmond, National Institutes of Health, Bethesda, MD) were used to model a distinct type of IRD[54]. *RPE65* mutations are associated with Leber congenital amaurosis type 2 (LCA2) and RP. The RPE65 enzyme is necessary for the functioning of the classical visual cycle, and a null mutation causes a full block of the cycle[73]. The *Rpe65*^-/-^ mice display complete cone dysfunction and rapid cone degeneration, whereas rod degeneration is relatively slow. The visual function in *Rpe65*^-/-^ is driven by residual rod-responses[45] although the source of chromophore is not known. The *Rpe65*^-/-^ mice were reared in vivarium conditions.

We also tested the TMB treatment in the *PDE6A*^-/-^ dog model of RP. The *PDE6A* mutation was a spontaneous mutation identified in the Cardigan Welsh Corgi breed. Affected dogs were bred with laboratory beagles to create the colony maintained at Michigan State University. The drug treatments were started at ~1-month of age, using subcutaneous infusion (see "*Drug administration*" below). Drug efficacy monitoring during the trial was based on photopic ERGs. At study termination at 7.5–8 months of age, the eyes were collected, and retinas processed as flat mounts for cone-population evaluation (see "*Retinal flat mounts, immunohistochemistry and microscopy*" below).

For all experimental procedures, animal subjects were treated in accordance with the NIH guidelines for the care and use of laboratory animals, the ARVO Statement for the Use of Animals in Ophthalmic and Vision Research, and the European Directive. Most mouse experiments were conducted at the University of California Irvine (UCI) where vivarium temperature ranged 21–24 °C, and humidity 30–75% (not controlled), and light-dark cycle was 12 hours (lights on 6 am to 6 pm). V1 electrophysiology experiments were conducted in Polish Academy of Science, Warsaw, Poland, and bulk RNA-seq analyzes at John Hopkins University. All protocols used in experiments on mice have been approved by the Institutional Animal Care and Use Committee (IACUC) at the UCI (protocol #AUP-21-096), or a permission was approved by the 1st Local Ethical Committee in Warsaw under the number 1400P2/2022. The dog experiments were conducted at Michigan State University (MSU) and were approved by the IACUC at MSU (# 05/14-090-00, # 05/17-075-00, and # PROTO202000013).

## Compounds, antibodies, and other reagents

Tamsulosin hydrochloride (T) and metoprolol tartrate (M) were purchased from TCI America (Portland, OR, USA). Bromocriptine mesylate (B) was purchased from LGM pharma (Erlanger, KY, USA) or Enzo Life Sciences (Farmingdale, NY, USA). Deuterated standards (±)-metopro-lol-d$_7$ and (R)-(-)-tamsulosin-d$_5$ for LC-MS/MS were purchased from CDN isotopes (Pointe-Claire, QC, Canada). α-Ergocryptine that was used as an internal standard for bromocriptine analyzes was purchased from Sigma-Aldrich (St. Louis, MO, USA). The drug diets were custom ordered from LabDiet (St. Louis, MO, USA), using the Prolab Isopro RMH3000 chow as base. PEG 200, kolliphor ELP, tetraglycol, and propylene glycol, used in the vehicle formulation for subcutaneous drug administration in dogs were all purchased from Sigma-Aldrich. The 4-hydroxynonenal (4-HNE) ELISA kit (ab238538) was purchased from Abcam (Waltham, MA, USA). Supplementary Table 1 presents sources of primary and secondary antibodies.

## Drug administration

Customized mouse diets incorporated into standard chow (Prolab IsoPro RMH3000, LabDiet) were used in most mouse studies, including T (0.05 mg/g,); M (2.5 mg/g); and B (0.25 mg/g). The utility of this dosing was confirmed by LC-MS/MS in the beginning of the study (see sections "*Blood and tissue collection for drug level quantification*" and "*Drug level quantification by LC-MS/MS*," below). In addition to the combination therapy using the three drugs (abbreviated as "TMB diet"), we tested the efficacy of each of the drugs alone, as well as the TM, TB, and MB drug pairs (dual therapies), in rd10 and *Rpe65*^-/-^ mice at the same dietary concentration as used in the TMB combination. In rd10 mice, we also tested a regimen where the concentrations of all three TMB compounds were lowered 5-fold (T 0.01 mg/g, M 0.5 mg/g, and B 0.05 mg/g) relative to the dose otherwise used throughout the study. The control groups received similar pellets (Prolab Isopro RMH3000, LabDiet) but without the drug(s).

The *PDE6A*^-/-^ dogs were treated *via* subcutaneous infusion using Alzet® osmotic infusion pumps (Cupertino, CA, USA). The pumps were filled with vehicle (25% PEG 200, 25% kolliphor ELP, 25% tetraglycol, 15% ethanol, 10% propylene glycol; Sigma-Aldrich) or drug in vehicle (T, 2.5 mg/ml; M, 20 mg/ml; B, 7.5 mg/ml). The pumps were surgically implanted under the skin over the chest or flank while the animals were under isoflurane anesthesia. The infusion pumps (Alzet model 2ML2) release the solution at a constant rate (5 μl/hr) for 2 weeks. Each pump was replaced with a new one every 14 days, and the total treatment duration was ~7-months. A total of 8 litters, between the years 2017–2020, were treated. Litters 1-5 (2017–2018) and litters 6–8 (2020) underwent somewhat different study designs, as follows. In studies with litters 1-5, the TMB-treated dogs received only one infusion pump at each surgery and control dogs remained untreated. In studies with litters 6-8, the TMB-treated dogs received one infusion pump per surgery for the two first months of the study; then (from the 5th surgery onwards) the dose was doubled by inserting two pumps simultaneously. Litter 6-8 control dogs received infusion pumps similarly but filled with vehicle alone.

## Blood and tissue collection for drug level quantification

Adult C57BL/6 J mice (RRID:IMSR_JAX:000664) were used for drug level testing during dietary TMB administration. Mice were acclimatized to the diet for at least 3 days prior to blood and tissue collection. Sample collection was performed as previously described[74]. Mice were terminally anesthetized by a 3- to 4-fold overdose of ketamine-xylazine cocktail (300-400 mg/kg ketamine; 30–40 mg/kg xylazine). When fully unresponsive, the mouse´s chest was cut open, a small hole was punctured in the right atrium of the heart, and ~0.5 ml of blood was quickly collected and transferred to a tube in wet ice. A perfusion needle was quickly inserted into the left ventricle, and the vasculature was perfused with ice-cold saline for 3 minutes using a peristaltic pump. After perfusion, the retinas were quickly excised by performing three incisions starting from the optic nerve head and cutting toward the *ora serrata*, which allowed easy and quick separation of the retina from the rest of the eye cup. Anterior parts of the eye were discarded, whereas the retinas and eyecups, consisting of the RPE, choroid, and sclera, were stored in 1.5 ml Eppendorf tubes and snap frozen. Blood was centrifuged at 4 °C and 17,000 g for 20 min, and thereafter the supernatant was collected as a serum sample. Blood collection from dogs was performed by venipuncture of the cephalic or jugular veins.

## Drug level quantification by LC-MS/MS

We conducted liquid chromatography-tandem mass spectrometry (LC-MS/MS) for quantification of drug levels in mouse and dog serum, and in extracts of mouse retina and eyecup ( = RPE-choroid--sclera) tissues. The mouse and dog subject #3 serum samples (100 µl) were precipitated with 400 µl of pre-cooled methanol, and centrifuged at 17,000 g for 15 min at 4 °C. The supernatant was carefully transferred to a SpinX centrifuge-tube filter with a 0.45 µm cellulose-acetate membrane (Costar, Salt Lake City, UT), and centrifuged at 7000 g for 2 min. The resulting filtrate was dried under vacuum. Eye cups or retinas from each mouse were homogenized in acetonitrile (2 × 800 µl). The resulting mixture was centrifuged at 17,000 g for 15 min at 4 °C. The supernatant was dried under vacuum. All of the dried samples were reconstituted in 100 µl of 50% methanol containing 100 ng/ml $d_7$-metoprolol, $d_5$-tamsulosin, and α-ergocryptine, and centrifuged at 17,000 g for 15 min at 4 °C. Twenty microliter of the supernatant was injected into an Ultimate 3000 HPLC system coupled with a LXQ mass spectrometer (ThermoFisher Scientific, Waltham, MA), with an electrospray ionization unit. The separation was performed on a Proshell EC-18 column (2.7 µm, 3.0 ×150 mm, Agilent, Santa Clara, CA), using a mobile phase consisting of 0.1% aqueous formic acid (A) and acetonitrile (B) at a flow rate of 200 µl·min⁻¹, with a gradient of B (15%-60% over 15 min). The capillary temperature was set at 375 °C. The signals were detected in positive multiple reaction mode (MRM) at conditions described in Supplementary Table 2 and quantified based on the standard curves representing the relationship between the amounts of authentic standards and the areas under the corresponding chromatographic peaks.

LC-MS/MS for serum samples from dog subjects 1-2 was ordered from the Proteomics & Metabolomics Core at Cleveland Clinic (Cleveland, OH, USA). Along with the calibration standard samples, 25 uL of serum from each of the 20 dog samples were spiked with 3 uL of 100 ng/mL internal standard, and protein was precipitated by adding 200 uL dry ice-cold methanol. These samples were incubated on ice for 30 min before centrifugation at 14,000 g for 5 min at 4 °C. The supernatants were carefully transferred to a SpinX 0.45 µm filter and centrifuged at 14,000 g for 2 min to remove any solid particles. Filtered samples were dried by vacuum centrifugation in a Speedvac. Each sample was reconstituted in 30 µL 80% methanol. The LC-MS/MS was performed by injecting 10 µL of each processed sample onto a Phenomenex 2 x 150 mm C18 column using a Vanquish UHPLC running at 0.30 mL/min with water and 0.1% formic acid as solvent A

and acetonitrile and 0.1% formic acid as solvent B. The mass spectrometer used was a Thermo TSQ Quantiva triple quadrupole running in positive ion mode. Each compound was monitored in a 2 min window at its retention time, and data acquisition was performed at a rate of 4 Hz.

## Drug efficacy testing in vivo by ERG and OCT

With mice, we tested retinal function by electroretinogram (ERG) recordings, and retinal structural changes by optical coherence tomography (OCT) in vivo, under anesthesia with ketamine (100 mg/kg) and xylazine (10 mg/kg). ERG recordings were performed with a Diagnosys Celeris ERG device (Lowell, MA, USA), as described earlier[63,74]. As we focused on cone protection in the RP context, we mainly recorded photopic ERGs, using rod-suppressing background light during light stimulation. Separate monochromatic UV (peak emission 370 nm, bandwidth ~50 nm) and green-light (peak emission 544 nm, bandwidth ~160 nm) stimulation were utilized to excite primarily mouse short-wavelength-sensitive (S[UV]) cones and medium-wavelength-sensitive (M) cones, respectively. The stimulation energy used with UV light was 509 mW/sr/m² and with green light 64 mW/sr/m² (10 and 30 cd·s/m² in Diagnosys Espion V6 software, respectively)[63]. Apart from one exception, mice were always kept in vivarium conditions and thus light-adapted prior to the photopic ERG recordings. The exception being that with dark-reared rd10 mice, the photopic ERG was recorded only once just prior to study termination and euthanasia, because dark-reared rd10 mice are highly susceptible to light damage even under normal laboratory light conditions[75]. The retinal function of dark-reared rd10 mice was monitored monthly, using a brief scotopic ERG protocol and OCT imaging in the dark room. Mice were handled and anesthetized in the dark room under dim red light. ERG stimulation was performed with two green flashes at 21 mW/sr/m² (10 cd·s/m² in Diagnosys Espion V6 software)[63] with an inter-stimulus interval of 25 sec. After the quick ERG protocol, OCT imaging was performed in a dim light environment with some background light coming from a fully dimmed computer monitor that was facing away from the imaging station. A standard scotopic ERG protocol[74] was performed with $Rho^{P23H/WT}$ mice at 3 months of age and with $Rpe65^{-/-}$ mice at 7 or 7.5 weeks of age. For this procedure, mice were fully dark-adapted overnight prior to recording. An additional 3 min were allowed in full dark adaptation after electrode insertion, before recordings were initiated. The stimulus intensities are depicted in the figures and legends. The ERG signal was acquired at 2 kHz and filtered with a low-frequency cutoff at 0.25 Hz and a high-frequency cutoff at 300 Hz. The Espion software automatically detected the amplitudes of the ERG a-wave (first negative ERG component) and b-wave (first positive ERG component); a-wave amplitude was measured from the signal baseline, whereas b-wave amplitude was measured as the difference between the negative trough (a-wave) and the highest positive peak.

OCT imaging in mice was performed with a Bioptigen spectral-domain OCT device (Leica Microsystems Inc., Buffalo Grove, IL), following previously published protocols[31]. The thickness of the outer nuclear layer (ONL; *i.e.* photoreceptor nuclei layer) is an established marker of rod photoreceptor degeneration in mice; therefore, its thickness was measured 500 µm away from the optic nerve head (ONH) in all retinal quadrants (nasal, temporal, superior and inferior), using ImageJ 1.47 v software (NIH, USA). We similarly assessed the magnitude of retinal detachment in OCT images. Values at each quadrant were averaged for each mouse and these averages were used for statistical analyzes. OCT image analyzes were performed by an experimenter blinded to treatment status.

In dogs, photopic ERGs were recorded under general anesthesia as previously described[76]. Briefly, puppies were anesthetized and maintained on isoflurane, pupils were dilated (Tropicamide Ophthalmic Solution UPS 1%, Falcon Pharmaceuticals Ltd., Fort Worth, TX,

USA), and the eyes were positioned in primary gaze using conjunctival positioning sutures. Jet-ERG corneal contact lens electrodes (Fabrinal Eye Care, La Chaux-De-Fonds, Switzerland) were used as active electrodes. Reference and ground electrodes were platinum skin electrodes (Grass Technologies, Warwick, RI, USA). An Espion E3 electrophysiology machine was used with a color-dome ganzfeld stimulator (Diagnosys LLC, Lowell, MA). A background white light of 30 cd/m$^2$ was used, and after 10 minutes of light-adaptation stimulation flashes of 3 cd·s/m$^2$ were applied. Responses to light-flashes were averaged for analysis.

## Visual cortex electrophysiology
Experiments were performed following our previously published protocols[63,77,78]. P41 mice (WT, $n = 6$ mice, $n = 21$ electrodes; rd10-vehicle, $n = 10$ mice, $n = 30$ electrodes; rd10-TMB, $n = 10$ mice, $n = 31$ electrodes) were initially anesthetized with 2% isoflurane in an O$_2$ stream (0.5 L/min), then placed into a stereotaxic apparatus. A small, custom-made plastic chamber was glued (Vetbond, St. Paul, MN) to the exposed skull. After one day of recovery, re-anesthetized animals were placed in a custom-made hammock, maintained under isoflurane anesthesia (1-2% in O$_2$), and multiple single tungsten electrodes were inserted into a small craniotomy above the visual cortex. Once the electrodes were inserted, the chamber was filled with sterile agar and sealed with sterile bone wax. During recording sessions, animals were kept under isoflurane anesthesia (0.5 – 1% in O$_2$). EEG and ECG were monitored throughout the experiments, and body temperature was maintained with a heating pad (in-house design).

Data were acquired using a 32-channel Scout recording system (Ripple, UT, USA). The local field potential (LFP) from multiple locations was bandpass filtered from 0.1 Hz to 250 Hz and stored with spiking data on a computer with a 1 kHz sampling rate. The LFP signal was cut according to stimulus time stamps and averaged across trials for each recording location to calculate VEPs[60]. The spike signal was bandpass filtered from 500 Hz to 7 kHz and stored in a computer hard drive at a 30 kHz sampling frequency. Spikes were sorted online in Trellis (Ripple, UT, USA) while performing visual stimulation. Visual stimuli were generated in Matlab 2021 (Mathworks, USA) using the Psychophysics Toolbox[79,80] version 3 and displayed on a gamma-corrected LCD monitor (Acer Predator Z35; 35 inches, 100 Hz; 3440 × 1440 pixels; 65 cd/m$^2$ mean luminance). Stimulus onset times were corrected for LCD-monitor delay using a photodiode and microcontroller (in-house design). For recordings of visually evoked responses, we used an ON-OFF stimulus where light (130 cd/m$^2$) turned on and off every 1500 ms for 100 times, as described in our previous work[60]. As the first step, evoked potentials across all layers were recorded, and the strongest response was used for comparisons between groups at the same cortical layer. The LFP signal was normalized using z-score standardization. The response amplitude of LFP was calculated as a difference between the peak of the positive and negative components in the VEP wave. The maximum of the response was defined as the maximum of either the negative or positive peak. The total number of single cells recorded in WT mice was 217; in vehicle-treated rd10 mice, 638; and in TMB-treated rd10 mice, 642 cells.

## Optomotor response test
The OMRs were assessed using a commercial OMR platform (Phenosys qOMR, PhenoSys GmbH, Berlin, Germany) that utilizes automated head tracking and behavior analysis, following previously published protocols[77]. The OMR arena was lit at ~80 lux, corresponding to the photopic light level. Spatial frequency tuning and visual acuity (VA) were tested with rd10 mice, and contrast sensitivity with $Rpe65^{-/-}$ mice. Rotating (12° sec$^{-1}$) vertical sinusoidal-grating stimuli at various spatial frequencies (SFs) were presented for 7 min per trial to light-adapted

rd10 mice after 3-4 weeks on treatment. Age-matched C57BL/6 J mice were used as healthy controls. The contrast between the white and black gratings was set at 100%, whereas the SF (0.1, 0.20, 0.25, 0.30, 0.35, 0.40, 0.45 cycles per degree of visual angle; CPD) pattern changed every 60 sec in a random order, with one exception; each session always started with 0.1 CPD to facilitate acclimatization to the task, as this SF is known to evoke reliable OMR in WT mice. Each mouse was tested in at least four trials (maximum two trials per day) and the first trial was considered acclimatization and not used in the analysis. The performances of the remaining trials were averaged for analysis, excluding those 60-sec stimulus periods that led to an OMR index (correct/incorrect ratio) smaller than 0.8.

In experiments with the $Rpe65^{-/-}$ mice, the spatial frequency of the grating was set at 0.1 CPD. This stimulus was presented at different contrast levels between the light and dark: 5, 10, 12.5, 15, 17.5, 20, 25, 50, and 100% contrast. Stimulation at each contrast level lasted 60 sec, and contrast levels were presented in a randomized order, except that each session was always started with 100% contrast to facilitate acclimatization to the task. Each trial in $Rpe65^{-/-}$ mice lasted for 9 min. Only male $Rpe65^{-/-}$ mouse data were used in the OMR test.

## Retinal flat mounts and immunohistochemistry
At drug-study termination, the mice were euthanized, the superior side of their eyes were marked, and their eyes were enucleated and processed for retinal flat-mount immunohistochemistry (IHC), as described before[32]. Retinal flat mounts were stained for cone-population inspection using anti-S- and anti-M-opsin antibodies (Supplementary Table 1).

After humane euthanasia, the dog eyes were processed for retinal flat mounting as previously described[60]. Peanut agglutinin (PNA) was applied (Supplementary Table 1) to bind to cone photoreceptors. AlexaFluor 488-conjugated streptavidin was used as secondary antibody.

## Fluorescence microscopy and cone counts
Retinal flat mounts were imaged with a fluorescence microscope (Keyence BZ-X800, Keyence, Itasca, IL, USA) equipped with an automated stage. To obtain retina panoramic images, the microscope was set to scan the whole dog or mouse retina flat mount with a 4x or 20x objective, respectively, using z-stacks. The z-dimensions were manually acquired in several parts of the retina before panoramic image acquisition. After acquisition, the z-stack was flattened, and all x-y sites were stitched to create a panoramic image.

Cone counting in experiments with vivarium-reared rd10 mice was computed using the hybrid cell count feature of the Keyence BZ-X800 Analyzer software, at default settings. Superior-inferior orientation of the retina was visually inspected from opposing S(UV)- and M-cone gradients in pigmented mice, wherein the S(UV)-cones are predominantly located in the inferior retina and the M-cones in the superior side[59]. For cell count analysis, circular sampling windows with 600-μm diameters were cropped from retinal panoramic images at the superior and inferior retina, centered at 500 μm from the ONH border.

Cone counting in experiments with the dark-reared rd10 mice and $Rho^{P23H/WT}$ mice was performed from higher resolution images. First, the whole retinal area was scanned, and a panoramic image acquired to permit accurate orientation. In mouse experiments, the objective was switched to a 40x magnification, and field of view was first set at the centerline with respect to the naso-temporal axis, and then moved inferiorly at 300, 750, and 1100 μm away from the ONH border, and then superiorly. Z-stacks were obtained covering the whole region of depth interest, using an imaging interval of 0.4 μm. In dogs, ONH appears slightly inferiorly and temporally from the midline, and the highest cone density area, area centralis (AC),

resides dorso-temporally from the ONH. Field of view was first set at the centerline with respect to the naso-temporal axis, and then using a 60x objective the imaging window was centered inferiorly at 1.5, 3.0, 4.5, and 6.0 mm away from the ONH border, and then superiorly at 1.5, 3.0, 4.5, 6.0, 7.5, 9.0, and 10.5 mm. Z-stacks were obtained covering the whole region of depth interest, using an imaging interval of $0.3\,\mu m$. Next, AC location was visually identified as being the highest cone density area, on the dorso-temporal side of the ONH. High resolution images were acquired, using a 100x objective, z-stack interval at $0.2\,\mu m$, and the optical sectioning tool of a Keyence BZ-X800 microscope. Maximum intensity projections were obtained from z-stacks. The images were black balanced to remove excess fluorescence background. Apart from AC images, counting from other images (40x and 60x) was computed using ImageJ software. From AC images, the PNA punctate (*i.e.*, cones) were manually counted. All imaging, computed cone-population analysis, and manual counting were performed by an experimenter blinded with regards to the treatments.

## Immunoblotting

Retinas were homogenized in $60\,\mu l$ of RIPA lysis buffer containing 25 mM Tris-HCl pH 7.6, 150 mM NaCl, 1% NP40, 1% Na deoxycholate, and 0.1% SDS, as well as benzonase nuclease (25 U per ml; Sigma-Aldrich) and protease inhibitor (cOmplete Mini EDTA-free, cat #11836170001 Roche Diagnostics, Mannheim, Germany). Immunoblotting was performed as described before[72]. Relative expression of proteins was evaluated by band-intensity analysis, using Image Studio Lite version 5.2 (Li-Cor, Lincoln, NE, USA); target protein expression was normalized to the expression level of α-tubulin. The antibodies used are presented in Supplementary Table 1.

## Enzyme-linked immunosorbent assay of 4-hydroxynonenal

Sex-matched rd10 and WT mice used for the experiment were dark-reared throughout their lifespan. Rd10 mice were treated with base diet or TMB diet between P28-P58. Age-matched WT mice were kept on a normal diet. At P58, mice were euthanized by cervical dislocation, eyes enucleated, and retinas were quickly dissected out. Both retinas were pooled, weighed as dry tissue samples, and homogenized in $110\,\mu l$ of ice-cold RIPA lysis buffer (25 mM Tris-HCl pH 7.6, 150 mM NaCl, 1% NP40, 1% Na deoxycholate, and 0.1% SDS) using a motorized plastic pestle. The homogenate was freshly used for determination of 4-hydroxynonenal (4-HNE) content in whole retina, as a duplicate ($50\,\mu l$ x 2). A commercially available 4-HNE ELISA kit (ab238538, Abcam) was used, as per manufacturer´s instructions.

## Ex vivo ERG

Mice were dark-adapted overnight and sacrificed by $CO_2$ asphyxiation. The whole retina was removed from each eyecup under infrared illumination and stored in oxygenated aqueous L15 (13.6 mg/ml, pH 7.4) solution (Sigma-Aldrich) containing 0.1% bovine serum albumin (BSA), at room temperature. The retina was oriented with the photoreceptor side up and placed on the filter paper attached to the dome of a perfusion chamber[46], between two electrodes connected to a differential amplifier. For transretinal ERG a-wave recordings, the tissue was perfused with bicarbonate-buffered Ames medium (Sigma-Aldrich) supplemented with 1.5–2 mM L-glutamate and $40\,\mu M$ DL-2-amino-4-phosphonobutyric acid (DL-AP4) to block postsynaptic components of the photoresponse, and with $100\,\mu M$ $BaCl_2$ to suppress the slow glial PIII component[46]. ERG b-wave recordings were performed in Ames medium supplemented with $100\,\mu M$ $BaCl_2$ only. The perfusion solution was continuously bubbled with a 95% $O_2$ / 5% $CO_2$ mixture and heated to 36–37 °C.

Photoreceptors were stimulated with 20-msec test flashes of calibrated 505-nm LED light. The light intensity was controlled by a computer in 0.5 log unit steps. Intensity-response relationships were fitted with the Naka-Rushton hyperbolic function:

$$R = \frac{R\max \cdot I^n}{I^n + I1/2^n} \tag{1}$$

where R is the transient-peak amplitude of the photoresponse, $R_{max}$ is the maximal response amplitude, I is the flash intensity, n is the Hill coefficient (exponent), and $I_{1/2}$ is the half-saturating light intensity. Photoresponses were amplified by a differential amplifier (DP-311, Warner Instruments), low-pass filtered at 300 Hz (8-pole Bessel), and digitized at 1 kHz. Data were analyzed with Clampfit 10.6 and Origin 8.5 software. Data from two retinas per mouse were averaged in cases where both retinas could be recorded: 11 retinas from 6 mice in the vehicle-group and 14 retinas from 7 mice in the TMB-group were used in the a-wave analysis; and 7 retinas from 6 mice in the vehicle-group and 7 retinas from 6 mice in the TMB-group were used in the b-wave analysis.

## Bulk-tissue RNA-seq

Before collection of retina samples from vivarium-reared rd10 mice (vehicle, $n = 8$; TMB, $n = 8$) for RNA-seq, the therapeutic effect of TMB treatment was tested using photopic ERG recording after 7 days on treatment (P36). The control groups, C57BL/6 J WT ($n = 4$) and young dark-reared rd10 mice ($n = 4$), were subjected to similar-lasting anesthesia at P36 and P27, respectively. Sexes were 50/50% balanced in all groups. On experimental day 8 (vivarium-reared rd10 and WT mice at P37; dark-reared rd10 mice at P28), the mice were euthanized by cervical dislocation, eyes enucleated and retinas promptly dissected. The retinas were homogenized using a motorized pestle, and the homogenate was spun in a tabletop centrifuge at full speed for 3 min. Supernatant was collected and RNA extracted using a Qiagen RNeasy Mini Kit (cat #74104; Qiaqen, Germantown, MD, USA) following the manufacturer's instructions. On-column DNAase digestion (Qiagen RNase-Free DNase; cat # 79254) was used to remove any genomic DNA contamination from the sample. RNA samples were sent to the Deep Sequencing and Microarray Core (Johns Hopkins University, USA) for quality control, library preparation, and sequencing. Briefly, poly-adenylated RNA was selected from the total RNA samples using Oligo-dT conjugated magnetic beads and prepared for sequencing according to the Illumina TruSeq RNA Sample Preparation Kit v2 (# RS-122-2001, Illumina). The libraries were sequenced for paired-end 75 cycles using the TruSeq SBS kit on the Illumina HiSeq system to obtain ~ 20–30 million reads per library.

The *Rpe65*[-/-] mice were treated with vehicle ($n = 4$; 2 females and 2 males) or TMB ($n = 5$; 3 females and 2 males) for 6 weeks (between P22-P64) before collection of retinas for RNA-seq analysis. Mice were anesthetized for ERG recording with ketamine (100 mg/kg) and xylazine (10 mg/kg) cocktail at P64 and euthanized by cervical dislocation under anesthesia. The age-matched C57BL/6 J WT mice ($n = 3$; 2 females and 1 male) were subjected to similar anesthesia, and euthanized for retina tissue collection. The sequencing analysis for this experiment was performed by NovoGene Co., Ltd. as previously described[72].

For analysis, the raw reads were pre-processed using FastQC v0.11.9 (https://www.bioinformatics.babraham.ac.uk/projects/fastqc/) to check the read quality. Trimmomatic v0.40[81] was used to remove low-quality reads and adapter contamination. The processed reads were then aligned to the reference genome GRCm38, using the STAR aligner 2.7.11b[82]. RseQC v5.0.1[83] was used to check the resulting aligned reads. The aligned reads were then assembled and quantified using HTSeq 2.0[84]. Differential gene expression analysis was performed using the DESeq2[85] package in R v4.2.1. The cutoff of significant differential expression was set at $P < 0.05$ after adjustment for multiple comparisons. Data for mitochondrial genes were extracted using bash script and run separately from chromosomal genes for differential expression analysis. After identifying the differentially expressed genes,

pathway analyzes were performed for gene annotation and functional enrichment analysis, using clusterProfiler v4.7.2 for R v4.2.1[86].

## Single-cell RNA-seq

Mice were euthanized, and eyes were enucleated for retinal tissue isolation. Retinal cells were dissociated using the Papain Dissociation System (Worthington Biochemical, Lakewood, NJ, USA) following the manufacturer's instructions, and diluted to a final concentration of 1000 cells/μl. For each rd10 mouse group, three retinas (one from each of $n = 3$ female mice) were pooled and used for single-cell RNA sequencing (scRNA-seq). The WT group consisted of two female and two male mice (total $n = 4$). Each group of *Rpe65*$^{-/-}$ mice consisted of $n = 7$ male mice, and the control WT group in this analysis consisted of $n = 3$ male mice. For each group, freshly dissociated cells (~16,500) were loaded into a 10× Genomics Chromium Single-Cell system using v3 chemistry, following the manufacturer's instructions. Libraries were pooled and sequenced on Illumina Nova-Seq6000 with ~500 million reads per library. The pre-processing steps such as generating and demultiplexing of FastQ files from raw sequencing reads (bclfastq, v2.20), aligning to UCSC mm10 transcriptome, and generating raw count matrices were conducted using Cell Ranger (v6.0.1) with default parameters. Cumulus software v1.5.0 was used to combine expression matrices[87] and to remove cell doublets by Scrublet v0.2.1[88]. The pipelines were run in Terra Cloud Platform (https://app.terra.bio/). Seurat 3.2.2[89] was used to perform downstream analysis following the standard pipeline using cells with ≥ 500 genes, resulting in the following number of analyzed cells. Rd10 mouse experiment: 4478 cells from WT mice; 5952 cells from vehicle-treated rd10 mice; and 4812 cells from TMB-treated rd10 mice. *Rpe65*$^{-/-}$ mouse experiment: 4135 cells from WT mice; 3979 cells from vehicle-treated *Rpe65*$^{-/-}$ mice; and 5901 cells from TMB-treated *Rpe65*$^{-/-}$ mice.

Principal Component (PC) analysis was performed on a submatrix of the top 1000 most variable genes computed using the function FindVariableGenes in Seurat. Batch effect between experimental conditions was minimized using the Harmony package v0.1.0 to remove non-cell-type-specific factors that may impact clustering[90]. The number of top PCs was assessed by the elbow method, keeping 19 PCs for clustering and data visualization. Cells were clustered using a shared nearest neighbor modularity optimization-based algorithm (FindClusters in Seurat). Cluster-specific genes were computed by FindAllMarkers in Seurat, using the MAST test with number of UMIs detected as a latent variable[91]. The same test was used to identify differentially expressed genes between experimental groups within cell clusters, which were annotated according to retinal cell-type-specific markers[78].

## Statistical analysis

Statistical analyzes were performed using GraphPad Prism (GPP) 10 software (La Jolla, CA, USA). The Shapiro-Wilk W test was used to test normal distribution in data that had group sizes smaller than 50, and the Kolmogorov-Smirnov normality test was used with larger datasets. The data sets where two groups were used to test a single variable were analyzed either by the parametric Welch's t-test (two-tailed), or with the Mann-Whitney u-test (two-tailed) if data was not continuous or normally distributed. If a single variable was compared in more than two groups, the one-way analysis of variance (ANOVA) test was used: ordinary ANOVA was used if standard deviations (SDs) were similar between groups, and Welch's ANOVA if SDs were different (assessed with the Brown-Forsythe test). One-way repeated measures (RM) ANOVA was used if the same animals were analyzed at different ages. However, if data was not Gaussian distributed, a non-parametric Kruskal-Wallis test was employed, followed by the Dunn's multiple comparisons test. Ordinary two-way ANOVA was applied to datasets that had two unmatching variables. RM-ANOVA

or mixed effects model (only applied in PNA puncta analysis due to two missing parameters) with Geisser-Greenhouse correction (no sphericity assumed) was used if factors matched. Ordinary, mixed effects model, and repeated measures ANOVAs were always followed by Bonferroni post hoc tests. Welch's ANOVA was followed by Dunnett's T3 tests. Data in the figures are presented as mean ± SD, while also showing values of all replicates when applicable. The threshold for statistical significance was set at $P < 0.05$. Outlier removal in the analysis of drug level in dog's blood was performed using the GPP's ROUT method with false discovery rate (Q) set at 0.5%.

## Reporting summary

Further information on research design is available in the Nature Portfolio Reporting Summary linked to this article.

## Data availability

The bulk RNA-seq and scRNA-seq data generated in this study have been deposited in the NCBI GEO database under accession code GSE238218. The raw and processed data generated in this study are provided in the Supplementary Information/Supplementary Data files/Source Data file. Source data are provided with this paper.

## Code availability

No custom code or mathematical algorithm was used. The paper does not report the original code.

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

## Acknowledgements

We thank Bryden J. Stanley, Emily Flanner, Billie Beckwith Cohen, Katelin Quantz, Jacquelyn Delvalle, Paige Winkler-Smith, and Kelian Sun for their help with experiments at Michigan State University. We thank the members and staff of the UCI Center for Translational Vision Research for technical assistance and helpful comments during this project. We thank Dr. Ling Li at the Cleveland Clinic Proteomics & Metabolomics Core unit for LC-MS/MS analysis. We also thank Drs. Zhongjie Fu and Lois Smith at Harvard University for invaluable critical comments at early phases of the project. This research was supported, in part, by grants from the National Eye Institute (NEI): EY034501 and EY009339 (K.P.), and the Department of Veterans Affairs: I01BX004939 (P.D.K.). H.L. was supported by the Knights Templar Eye Foundation Career-Starter Research Grant, Fight for Sight Postdoctoral Award, Finnish Cultural Foundation, The Osk. Huttunen Foundation, Orion Research Foundation, Finnish Eye and Tissue Bank Foundation, Finnish Retina Registered Association, and Sokeain Ystävät/De Blindas Vänner Registered Association, and the Academy of Finland (#346295). This work was made possible, in part, through access to the Genomics High-Throughput Facility Shared Resource of the Cancer Center Support Grant (P30CA062203) at the University of California, Irvine (UCI). The authors

acknowledge support through the Gavin Herbert Eye Institute at UCI from an unrestricted grant from Research to Prevent Blindness and from NIH core grant P30EY034070. This work was also partially supported by the National Science Center, Poland Grant 2019/34/E/NZ5/00434 (to A.T.F.); The International Centre for Translational Eye Research (MAB/2019/12) project is carried out within the International Research Agendas programme of the Foundation for Polish Science co-financed by the European Union under the European Regional Development Fund.

## Author contributions

H.L. designed the research; H.L., J.Z., L.M.O., E.H.C., L.F.M., J.Q., A.V.K., A.G., K.K., T.H., D.L., T.T.L., Z.D., S.P-J. conducted experiments; H.L., J.Z., U.S., A.V.K., T.T.L., E.E.E., D.E.E., M.T., A.F., S.P-J. performed data analyzes; H.L. wrote the manuscript; P.D.K., K.P. contributed to the written manuscript; H.L., J.Z., L.M.O., E.H.C., A.V.K., V.J.K., A.F., S.P-J., K.P. contributed to methodology; H.L., S.B., A.F., V.J.K., S.P-J., K.P. contributed to resources and funding acquisition; H.L., P.D.K., S.B., V.J.K., A.F., S.P-J. K.P. provided project supervision. All authors have reviewed and approved the final version of the manuscript.

## Competing interests

The authors declare the following competing interests: K.P. is a consultant for Polgenix Inc. and serves on the Scientific Advisory Board at Hyperion Eye Ltd. All other authors declare no competing interests.

## Additional information

[1]School of Pharmacy, Faculty of Health Sciences, University of Eastern Finland, Yliopistonranta 1C, 70211 Kuopio, Finland. [2]Gavin Herbert Eye Institute-Center for Translational Vision Research, Department of Ophthalmology, University of California, Irvine, CA 92697, USA. [3]Small Animal Clinical Sciences, Michigan State University, East Lansing, MI 48824, USA. [4]International Centre for Translational Eye Research, Warsaw, Poland. [5]Institute of Physical Chemistry, Polish Academy of Sciences, Warsaw, Poland. [6]Department of Ophthalmology, Department of Cell & Developmental Biology, Ann Arbor, MI 48105, USA. [7]Department of Physiology and Biophysics, School of Medicine, University of California - Irvine, Irvine, CA 92697, USA. [8]Department of Clinical Pharmacy Practice, School of Pharmacy and Pharmaceutical Sciences, University of California - Irvine, Irvine, CA 92697, USA. [9]Research Service, VA Long Beach Healthcare System, Long Beach, California 90822, USA. [10]Department of Ophthalmology, Johns Hopkins University School of Medicine, Baltimore, MD 21205, USA. [11]Department of Neurology, Johns Hopkins University School of Medicine, Baltimore, MD 21205, USA. [12]Institute for Cell Engineering, Johns Hopkins University School of Medicine, Baltimore, MD 21205, USA. [13]Kavli Neuroscience Discovery Institute, Johns Hopkins University School of Medicine, Baltimore, MD 21205, USA. [14]Solomon H. Snyder Department of Neuroscience, Johns Hopkins University School of Medicine, Baltimore, MD 21205, USA. [15]Department of Chemistry, University of California-Irvine, Irvine, CA 92697, USA. [16]Department of Molecular Biology and Biochemistry, University of California-Irvine, Irvine, CA 92697, USA. [17]Present address: Gavin Herbert Eye Institute-Center for Translational Vision Research, Department of Ophthalmology, University of California, Irvine, CA 92697, USA. ✉e-mail: henri.leinonen@uef.fi; kpalczew@uci.edu

