## [Peer Review File · Nature Communications]

REVIEWER COMMENTS

Reviewer #1 (Remarks to the Author):

The manuscript reports on an incredible amount of studies and experiments across two mouse and one dog model.

Main comments:

1. The rationale underlying the selection of tamsulosin, metoprolol and bromocriptine did not become fully clear to me from the Introduction in multiple ways: Firstly, if the authors wish to target cAMP and Ca pathways, are alpha-1A, beta-1 and dopamine D2 receptors the most relevant targets controlling this in these pathways in the (human) retina? Second, why have the specific drugs been chosen as representatives of their classes? For instance, tamsulosin has drawn criticism because of possible induction of IFIS, a possible complication of cataract surgery that may occur less often with other alpha-blockers. Moreover, why choose a moderately alpha-1A-selective drug and not one also blocking the other subtypes? Also, metoprolol has moderate selectivity for beta-1 over beta-2 adrenoceptors. Why not choose one that has greater selectivity for beta-1 (e.g., bisoprolol) or one that hits both beta-1 and beta-2 receptors (e.g., propranolol)? Within each drug class compounds with more favorable pharmacokinetic profiles may be available. Or were the three drugs mostly chosen because many ophthalmologists are familiar with them? Bottom line, the underlying rationale at the receptor and compound levels needs better explanation.
2. The Abstract would be more informative by including some indications of effect sizes. For instance, it makes a big difference whether degeneration in the mouse models was slowed by 2 days or 2 years. Actually, if I look at e.g., Figure 4GIJ (time course over months 2-8) the attenuation of progression is visible but less impressive than that of some of the single time point measurements. Therefore, I would appreciate a more honest description of how much slowing of progression actually is achieved.
3. Professional statisticians discourage reporting p-values in the absence of effect size indicators (e.g., l. 189, 195, 197 and so forth). Why not simply at least semi-qualitatively describe the effect size here and move the p-value to the figure or figure legend?
4. While you have an interesting biological hypothesis, the overall study appears to me as exploratory in a statistical sense (i.e., no quantifiable statistical null hypothesis and pre-study power calculations evident from the manuscript). I have no problem. However, it follows from this that the calculated p-values cannot be interpreted as hypothesis-testing and only as descriptive. This should be stated explicitly.
5. SEM is an indicator of precision, not of variability; as precision indicator it is less informative than 95% confidence intervals and makes more assumption. In the figures error bars should describe variability, and based on recommendations from leading statisticians this should be done by SD (not by SEM) [1]. Please convert accordingly.

6. When looking at some of the figures, specifically the box & whisker plots (fine with me), it appears that you are not assuming normal populations. If that is the case, consequently you should apply non-parametric testing to those experiments. The overall description for which parameter you assume a normal distribution (and apply parametric tests) and for which you do not assume this and apply non-parametric tests remains too vague to me.

7. It remains unclear to me to what the n refers in text and figures. Is it each datapoint one animal, one eye, or one cell? All analyses should be based on biological, not on technical replicates [2].

Other comments:

8. The way you put the drugs into the food pellets would be equivalent to an immediate-release drug formulation (capsule). Please note that at least for tamsulosin a modified release formulation is used in the US (this and additional other modified release formulations are available in Canada, Asia, and Europe). I understand the rationale of measuring one time point for drug concentration only on pragmatic grounds of animal experiments. However, this single time point may deviate from typical exposures (not just C_{max}) in people. That may be worth mentioning.

9. L. 653-656: I could not follow this conclusion. Use of these drugs in their approved indications will only very rarely lead to use of this triple combination. On the other hand, monotherapies and some of the dual treatments (at least in mice) were not effective according to the present data. Thus, this speculation is interesting but in its present wording misleading.

10. Please add RRID wherever feasible, e.g., for animal strains and antibodies. If unfeasible, please add catalog numbers for unequivocal identification of resources.

11. Reporting on the animals does not include the minimum information specified in the ARRIVE 2.0 guidelines, e.g., related to sex and body weight at start [3]. Moreover, if mixed groups of male and female animals were used, the specific distribution should be made transparent in each figure legend.

12. Given the multitude of participating institutions, it may be helpful to indicate in greater detail which data were generated at which institution.

13. Microscopy experiments are notoriously vulnerable to investigator bias. Please state explicitly whether the person performing the experiments e.g., selecting the viewing fields was blinded to group allocation.

14. L. 1025: Whether to use parametric or non-parametric tests depends on assumption of presence of normality in the underlying populations (not in the samples). The question of comparable variances across groups is distinct from this. For instance, versions of the parametric t-test and many other parametric tests are available that do not assume equal variance. Please explain.

15. Each figure legend must include a specific sample size for each group.

References

1. Michel MC, Murphy TJ, Motulsky HJ. New author guidelines for displaying data and reporting data analysis and statistical methods in experimental biology. *Mol Pharmacol.* 2020;97(1):49-60.

2. Eisner DA. Pseudoreplication in physiology: more means less. *J Gen Physiol.* 2021;153(2):e202012826.
3. Percie du Sert N, Hurst V, Ahluwalia A, Alam S, Avey MT, Baker M, et al. The ARRIVE guidelines 2.0: updated guidelines for reporting animal research. *Br J Pharmacol.* 2020;177(16):3617-24.

Reviewer #2 (Remarks to the Author):

This is an extraordinarily comprehensive study of the potential for catecholaminergic GPCR drug combinations to serve as disease against RD. The drug combination is quite effective for slowing rd10 degeneration and less effective at slowing degeneration caused by rhodopsin and RPE65 mutations. The authors somewhat overstate its effectiveness for the rhodopsin and RPE65 induced degenerations, but the rd10 data are very impressive. The analyses are thorough and they provide lots of useful information about changes in transcription, oxidative damage and responsiveness in various cell types in retinas that occur during and in response to degeneration and in response to the drugs (either individually or in combination).

I have only minor comments and suggestions.

1. Fig. 1B is confusing. Is the y-axis the number of micromoles in all of the blood in the animal? Or is it per microliter or some other unit? The same ambiguity for retina and eyecup needs to be clarified.
2. I'm pretty sure panels 1D-G are from mice a week after moving to the vivarium but that information should be stated directly in the figure legend.
3. Related to the ERG results in Fig. 1 - was there any effect on scotopic responses?
4. What does the "CAT" arrow on the upper right side of the lower panel of Fig. 2K refer to - there does not seem to be visible band there.
5. line 563: "where" should be "were".

6. line 567: The observation of massive changes in MG transcription is an important finding of this study. Has this been reported before? If not the authors should emphasize that this is a novel finding (or it has already been shown then the authors should cite the previous studies.)

7. Line 567-8: Is it likely that the TMB drugs are working directly on the Müller cells or is the effect indirect? It may not be possible to answer this definitively, but it is an interesting question and should be discussed.

Reviewer #3 (Remarks to the Author):

This research contributes to an important area of research of mutation agnostic, pharmacologic therapeutics for inherited retinal dystrophies.

This work expands upon previous work by the authors in examining the therapeutic potential of GPCRs, tamsulosin, metoprolol and bromocriptine in additional models of IRD.

The research was noteworthy in that it established the utility of TMB combo therapy useful in several murine models and a canine model of IRD all with varying pathophysiology of disease.

In presenting the data regarding adrenergic and dopaminergic signaling in Rd10 mice (lines 242-244, Figure 1M), representation of results were unclear with current figure and text as to whether there were any significant differences between Veh treated and TMB treated Rd10 mice; it appears that comparisons were made between dark-reared mice and these two groups but not between veh and treatment group. Clarification and expanding upon this section of the results is especially important given the presumed mechanism in these catecholaminergic GPCR drugs used in this study, and the presumed affect on the pathology in these IRD models as touched upon in the discussion (lines 623-656).

In assessing changes in M and S(UV) cones in Rd10 treated mice (Fig 1U-Z), it appears that while M cones were improved, only one outlier mouse appears to have an increase in both S cone count and ERG. Given discussion of results on lines 255-260. Would recommend additional n's to clarify this finding.

In evaluating changes in RPE65^{-/-} mice, there are presumed largely inner retinal changes given the ERG findings. It would have been helpful to also conduct similar scRNA-seq experiment to provide greater detail on changes occurring on which inner retinal cells, as was performed and discussed on dark-reared Rd10 mice.

Lastly, studies on the Pde6a^{-/-} canine model showed less of an effect, possibly largely due to administration and dosing/undertreatment. Assessing drug concentration levels in eyecup, even on WT or control dogs, would be helpful to elucidate whether this was the main limiting factor, or whether lack of effect may be due to either mechanistic/pathophysiology differences in a different IRD model or inherent differences in species and GPCR signaling.

Other than the above recommendations on data interpretation and findings, data interpretation and methodology of the rest of the experiments presented were sound and contributes to the important search of additional, more practical therapeutics for IRDs.

Reviewer #1 Comments

The manuscript reports on an incredible amount of studies and experiments across two mouse and one dog model.

Main comments:

1. The rationale underlying the selection of tamsulosin, metoprolol and bromocriptine did not become fully clear to me from the Introduction in multiple ways: Firstly, if the authors wish to target cAMP and Ca pathways, are alpha-1A, beta-1 and dopamine D2 receptors the most relevant targets controlling this in these pathways in the (human) retina? Second, why have the specific drugs been chosen as representatives of their classes? For instance, tamsulosin has drawn criticism because of possible induction of IFIS, a possible complication of cataract surgery that may occur less often with other alpha-blockers. Moreover, why choose a moderately alpha-1A-selective drug and not one also blocking the other subtypes? Also, metoprolol has moderate selectivity for beta-1 over beta-2 adrenoceptors. Why not choose one that has greater selectivity for beta-1 (e.g., bisoprolol) or one that hits both beta-1 and beta-2 receptors (e.g., propranolol)? Within each drug class compounds with more favorable pharmacokinetic profiles may be available. Or were the three drugs mostly chosen because many ophthalmologists are familiar with them? Bottom line, the underlying rationale at the receptor and compound levels needs better explanation.

The expression of target receptor transcripts in mouse and human retinas was demonstrated previously (Table 1 in Chen et al. 2016). Lowered cAMP production could of course be achieved with e.g., selective adenylyl cyclase inhibitors. However, such therapy administered via systemic delivery would likely not be well tolerated, and there are no clinically approved agents in this class. Intracellular Ca²⁺ could be targeted by repurposing approved calcium-channel blockers (Barabas, Cutler Peck, and Krizaj 2010), but the therapeutic effect of such interventions in RD is likely modest and/or transient (Kilicarlsan et al. 2021). GPCRs are highly druggable (1/3 of all approved drugs affect GPCRs) and targeting them typically has excellent safety profile.

Reviewer #1 is raising an important point about drug choice. We modified the discussion chapter and added that also other alpha- and beta-blockers or D2-like agonists could be applicable in the context. First/second sentence of introduction now: *“In the current study, we utilized a Systems Pharmacology-based approach and targeted retina-expressing G_q-, G_s- and G_i-coupled GPCRs (Chen et al. 2016) simultaneously using clinically approved drugs tamsulosin (T), metoprolol (M), and bromocriptine (B) as example compounds. Also other alpha- and beta-receptor blockers (Chen et al. 2016; Kern et al. 2021; Kanan et al. 2019) and D2-like receptor agonists (Shibagaki et al. 2015) have shown protective properties against RD, and could potentially be used in the context.”*

Due to word count constraints, our ability to add text to the manuscript is limited. However, to further clarify for reviewers, we have tested, among others, doxazosin (Chen et al. 2016), bisoprolol (Fig. 1 below), and pramipexole (Kern et al. 2021), as alternative alpha-1-blockers, beta-blockers, and D2-agonists, respectively. Doxazosin and bisoprolol could work as well in the context of the current study. Pramipexole was not effective in diabetic retinopathy model (Kern et al. 2021), however, we should note that it is much more potent at D3 compared to D2. The D3 receptor is not expressed in murine nor in human retina.

Figure 1. Selective beta-1-receptor blockers protect against light-induced retinal degeneration (LIRD) in ABCA4/RDH8 DKO mice. Optical coherence tomography (OCT) was performed 7 days after LIRD induction and outer nuclear layer (ONL) thickness was measured from OCT images.

As assumed by the reviewer, we did have some practical reasons too for the drug choices. Metoprolol is the most prescribed beta-blocker drug in the USA (<https://www.definitivehc.com/blog/beta-blocker-prescription-patterns>). In addition, metoprolol displays high melanin-binding (Pitkänen et al. 2007) which could target it better to the RPE/choroid complex highly enriched in melanin. Tamsulosin is also the most highly utilized drug in its primary clinical indication (benign prostate hyperplasia), and it has a low propensity for clinical side effects (Korstanje, Krauwinkel, and van Doesum-Wolters 2011). We are aware of tamsulosin’s association with intra-operative floppy iris syndrome (IFIS), but the systemic safety counterbalances this. Also, IFIS is now well recognized and can be handled by alternative surgical techniques. Nevertheless, the optimal alpha-1-blockers, beta-blockers, and D2-agonists should be systemically studied, but this is a very laborious task which will last for several years. The first author is currently leading a 5-year program (Research Council of Finland grant # 346295) to address these issues.

2. The Abstract would be more informative by including some indications of effect sizes. For instance, it makes a big difference whether degeneration in the mouse models was slowed by 2 days or 2 years. Actually, if I look at e.g., Figure 4GIJ (time course over months 2-8) the attenuation of progression is visible but less impressive than that of some of the single time point measurements. Therefore, I would appreciate a more honest description of how much slowing of progression actually is achieved.

Based on the reviewer’s comments, we modified the abstract as follows: “Cone degeneration was also modestly mitigated after a 7-month-long TMB infusion in PDE6A-/- dogs. In the Rpe65-/- mouse model of Leber congenital amaurosis, dietary TMB improved rod-pathway function and visual behavior but did not protect from cone degeneration.” Although we agree with the reviewer about the utility of including effect sizes, word limits prevent us from including them in the abstract. However, to address this comment and the next (#3), we incorporated statements about effect sizes throughout the results section. These additions are visible as track-changes in the revised manuscript.

3. Professional statisticians discourage reporting p-values in the absence of effect size indicators (e.g., l. 189, 195, 197 and so forth). Why not simply at least semi-qualitatively describe the effect size here and move the p-value to the figure or figure legend?

We have added effect size indicators to relevant parts to make the main text more intelligible. We cannot put results of statistical tests to figure legends due to limited word counts (<350), but we include asterisks in the figures itself. Results of all statistical tests are also included in the source data excel files, required by the journal.

4. While you have an interesting biological hypothesis, the overall study appears to me as exploratory in a statistical sense (i.e., no quantifiable statistical null hypothesis and pre-study power calculations evident from the manuscript). I have no problem. However, it follows from this that the calculated p-values cannot be interpreted as hypothesis-testing and only as descriptive. This should be stated explicitly.

We agree with the reviewer. From a statistical point of view, the study is exploratory, as are practically all preclinical studies. Please note the breadth of investigation here that lasted five years starting from early 2017. We added the word “exploratory” in the relevant part in the results section: *“To begin exploratory chronic drug trials in mouse models (Fig. 1A), we needed to establish a suitable method of drug administration.”*

5. SEM is an indicator of precision, not of variability; as precision indicator it is less informative than 95% confidence intervals and makes more assumption. In the figures error bars should describe variability, and based on recommendations from leading statisticians this should be done by SD (not by SEM) [1]. Please convert accordingly.

We have modified all data presentation and changed SEMs to SDs.

6. When looking at some of the figures, specifically the box & whisker plots (fine with me), it appears that you are not assuming normal populations. If that is the case, consequently you should apply non-parametric testing to those experiments. The overall description for which parameter you assume a normal distribution (and apply parametric tests) and for which you do not assume this and apply non-parametric tests remains too vague to me.

Normal distribution was tested with Shapiro-Wilk W test (small sample sizes), or with Kolmogorov-Smirnov test if sample size >50. We have clarified these points in the figure legends and “statistical analysis” section of the Materials and Methods. We also used non-parametric tests on occasions

when data was not continuous (e.g. ONL thickness or retinal detachment which had several 0 parameters). Results of all statistical tests are also included in the source data excel files.

7. It remains unclear to me to what the n refers in text and figures. Is it each datapoint one animal, one eye, or one cell? All analyses should be based on biological, not on technical replicates [2].

We clarified this in Figure 1 and 3 legends. Analyzing the total number of recorded cells is standard in single-cell electrophysiology. Similarly, using each single cell as a replicate is standard in scRNA-seq. High number of biological replicates is not feasible using these methods. For instance, running one single cell suspension for scRNA-seq still costs thousands of dollars. We do agree though that the number of animals (biological replicates) where data originates from must be clearly stated.

Other comments:

8. The way you put the drugs into the food pellets would be equivalent to an immediate-release drug formulation (capsule). Please note that at least for tamsulosin a modified release formulation is used in the US (this and additional other modified release formulations are available in Canada, Asia, and Europe). I understand the rationale of measuring one time point for drug concentration only on pragmatic grounds of animal experiments. However, this single time point may deviate from typical exposures (not just Cmax) in people. That may be worth mentioning.

Absolutely. We added the following texts in early results section and discussion, respectively:

In results: *“Overall, dietary TMB administration proved to be applicable for drug-efficacy testing in chronic mouse RD paradigms, although serum level variance with the method is large and the drug levels tested at single time points in mice are not directly comparable to typical exposure patterns in humans.”*

In discussion: *“Our goal was to attain sustained retinal protective effect in vivo in several etiologically distinct chronic RD models via dietary (in mice) or subcutaneous infusion (in dogs) administration. Despite the heterogeneity of the disease models used in this study, and the inability to precisely control dosing, TMB combination treatment was effective in all the tested paradigms.”*

We do not see the inability to precisely control dosing in mice as such a negative point though since the drug levels will vary between-subjects, and even within-subjects (compliance, administration time, eating...) also in clinical use. We see that the therapeutic strategy is more likely to work outside-of-laboratory-control since it worked even in such unstable paradigm as dietary drug administration in mice is.

For reviewers interest, we did test many different kinds of administration routes before starting the dietary experiments in mice, including administration in drinking water (drugs were not chemically stable + drug taste comes through in simple water), infusion with Alzet pumps (stressful for mice + lower drug

serum concentrations achieved), and bolus injections or gavage (does not work at all since half-life of elimination of T, M and B very short in mice + excessively stressful in long experiments).

9. L. 653-656: I could not follow this conclusion. Use of these drugs in their approved indications will only very rarely lead to use of this triple combination. On the other hand, monotherapies and some of the dual treatments (at least in mice) were not effective according to the present data. Thus, this speculation is interesting but in its present wording misleading.

The reviewer is right. Clinical situations where all these three are in use are ultrarare. It would take so much space to explain this properly that we decided to remove the whole paragraph.

10. Please add RRID wherever feasible, e.g., for animal strains and antibodies. If unfeasible, please add catalog numbers for unequivocal identification of resources.

Supplementary table 1 shows sources and catalog numbers of all antibodies used. RRIDs of commercially available animal models were added. Sources of drugs and chemicals are provided.

11. Reporting on the animals does not include the minimum information specified in the ARRIVE 2.0 guidelines, e.g., related to sex and body weight at start [3]. Moreover, if mixed groups of male and female animals were used, the specific distribution should be made transparent in each figure legend.

The journal requires a reporting summary and source data file(s) that will be published together with the article, which obliges us to follow the requirements of ARRIVE guidelines. With respect to the body weight, we verified that dietary TMB did not affect weight gain up to 7-month of age in mice (supplementary figure 1). We tested body weight in subset of the experiments, but not all, as it became apparent that there is no negative effect. With respect to the sex distribution in each analysis, this is unfortunately infeasible to include into the main text due to word count limitation (<350 words in figure legends). However, all this information (plus results of statistical tests) is included in the source data files.

12. Given the multitude of participating institutions, it may be helpful to indicate in greater detail which data were generated at which institution.

We modified the relevant section so that it would be clearer: *“For all experimental procedures, animal subjects were treated in accordance with the NIH guidelines for the care and use of laboratory animals, the ARVO Statement for the Use of Animals in Ophthalmic and Vision Research, and European Directive. Most mouse experiments were conducted at the University of California Irvine (UCI). V1 electrophysiology experiments were conducted in Polish Academy of Science, Warsaw, Poland, and bulk RNA-seq at John Hopkins University.”*

All protocols used in experiments on mice have been approved by the Institutional Animal Care and Use Committee (IACUC) at the University of California Irvine (UCI), or a permission was approved by the 1st Local Ethical Committee in Warsaw under the number 1400P2/2022. The dog experiments were conducted at Michigan State University (MSU) and were approved by the IACUC at MSU.”

13. Microscopy experiments are notoriously vulnerable to investigator bias. Please state explicitly whether the person performing the experiments e.g., selecting the viewing fields was blinded to group allocation.

Following standard procedures, OCT imaging was always centered to the optic nerve head. A student in the lab performed the image analyses, and this is mentioned in the manuscript: “*OCT image analyses were performed by an experimenter blinded with regards to the treatments*”.

The location of imaging fields for cone opsin and PNA counts was predetermined and students in the lab were instructed to perform imaging consistently across subjects. Both microscopy and image analysis were performed by experimenters blinded with regards to the treatments. This is/was also mentioned in the manuscript.

14. L. 1025: Whether to use parametric or non-parametric tests depends on assumption of presence of normality in the underlying populations (not in the samples). The question of comparable variances across groups is distinct from this. For instance, versions of the parametric t-test and many other parametric tests are available that do not assume equal variance. Please explain.

This is now clarified in the *statistical analysis* section. We changed standard t-tests to Welch’s t-tests throughout the manuscript. We performed normality tests (Shapiro-Wilk), and used nonparametric tests when data was significantly non-normally distributed. Nonparametric tests were also used when data was not continuous such sometimes seen in OCT thickness or retinal detachment (many 0 values). Results of statistical tests are included in the data source file(s).

15. Each figure legend must include a specific sample size for each group.

Unfortunately, this is not practical due to the constraining word count limit and large amount of data in each figure. We have used scatter plots so that readers can see at least an estimate if not actual numbers of the sample sizes. All data is made transparent in the data source file(s) that is published together with main text and supplementary material.

References

1. Michel MC, Murphy TJ, Motulsky HJ. New author guidelines for displaying data and reporting data analysis and statistical methods in experimental biology. *Mol Pharmacol.* 2020;97(1):49-60.

2. Eisner DA. Pseudoreplication in physiology: more means less. *J Gen Physiol.* 2021;153(2):e202012826.
3. Percie du Sert N, Hurst V, Ahluwalia A, Alam S, Avey MT, Baker M, et al. The ARRIVE guidelines 2.0: updated guidelines for reporting animal research. *Br J Pharmacol.* 2020;177(16):3617-24.

Thank you for these references.

Reviewer #2 Comments

This is an extraordinarily comprehensive study of the potential for catecholaminergic GPCR drug combinations to serve as disease against RD. The drug combination is quite effective for slowing rd10 degeneration and less effective at slowing degeneration caused by rhodopsin and RPE65 mutations. The authors somewhat overstate its effectiveness for the rhodopsin and RPE65 induced degenerations, but the rd10 data are very impressive. The analyses are thorough and they provide lots of useful information about changes in transcription, oxidative damage and responsiveness in various cell types in retinas that occur during and in response to degeneration and in response to the drugs (either individually or in combination).

Thank you for this great summary.

I have only minor comments and suggestions.

1. Fig. 1B is confusing. Is the y-axis the number of micromoles in all of the blood in the animal? Or is it per microliter or some other unit? The same ambiguity for retina and eyecup needs to be clarified.

Thank you for pointing out this ambiguity. The units were originally presented as moles per mg of serum or tissue. We changed this to mass units (ng/mg of serum or tissue, which is not shown in Figure 1B) so that it may make intuitively more sense with other parts that show only serum levels (presented as ng/ml of serum).

2. I'm pretty sure panels 1D-G are from mice a week after moving to the vivarium but that information should be stated directly in the figure legend.

There's a heading above the panels stating that it is 1 week after. We made it clearer. We need to be minimalistic in the figure legend due to word count limitations (<350).

3. Related to the ERG results in Fig. 1 - was there any effect on scotopic responses?

In this paradigm, we did not measure scotopic ERGs. The rods die very quickly in Rd10 mice when reared in vivarium. The scotopic ERGs too then primarily reflect cone-mediated responses.

4. What the "CAT" arrow on the upper right side of the lower panel of Fig. 2K refer to - there does not seem to be visible band there.

The same membranes were incubated in the anti-GFAP antibody, that had been first incubated in anti-CAT antibody. Therefore, the CAT signal at ~60 kDa stays. It is barely visible in the lowest panel in Fig.

2K because if we increase exposure, then GFAP signal saturates. For revision, we decreased exposure slightly in the lowest panel to show only GFAP, and removed the arrowhead and CAT-text to avoid confusion.

5. line 563: "where" should be "were"

Thanks for pointing out this typo. Corrected.

6. line 567: The observation of massive changes in MG transcription is an important finding of this study. Has this been reported before? If not the authors should emphasize that this is a novel finding (or it has already been shown then the authors should cite the previous studies.)

Thanks for the comment. Due to word and citation count constraints set by the journal, we cannot discuss this extensively. But we added brief discussion about this to lines 585-591: *“Due to rod death in rd10 mice, the largest cell cluster identified in these mice was the Müller glial cells, the main glia cells in the retina which have a crucial role in maintaining metabolic support, homeostasis, and tissue integrity. MGs are known to respond robustly to retinal stress, and this response can be either protective or detrimental to retinal function. Indeed, MGs showed massive differences in transcriptomic regulation between mutant and WT mice in our analyses. TMB significantly affected the expression of almost 500 genes in the MGs of rd10 mice, and almost 1000 genes in the MGs of Rpe65^{-/-} mice.”*

7. Line 567-8: Is it likely that the TMB drugs are working directly on the Müller cells or is the effect indirect? It may not be possible to answer this definitively, but it is an interesting question and should be discussed.

Based on the scRNA-seq results this seems like a viable option. Most DE genes were found in the Müller cells when vehicle- and TMB-groups were compared in both rd10 (Fig. 3) and *Rpe65^{-/-}* (Fig. S10) mouse datasets. Without direct evidence, we abstain speculating about this in the manuscript, and we want to leave the readers some room for their own conclusions. Rather, we state the facts in the discussion as follows (lines 591-598): *“In the rd10 model, marked interconnectedness of the downregulated genes was observed, as most of the genes were connected through a single network, which was largely associated with decreased cellular response to stress. In contrast, TMB-mediated upregulated genes in the MGs of rd10 mice were organized into five distinct functional networks that were associated with oxidative phosphorylation, ribosome, core histone, circadian rhythm, and photoreceptor outer segment/heterotrimeric G-protein complex. In Rpe65^{-/-} mice, TMB treatment enhanced mitochondrial gene expression and metabolism-associated gene sets in rod, RBC and MG clusters, which may explain the improved visual function as detected by ERGs and OMR.”*

Reviewer #3 Comments

This research contributes to an important area of research of mutation agnostic, pharmacologic therapeutics for inherited retinal dystrophies.

This work expands upon previous work by the authors in examining the therapeutic potential of GPCRs, tamsulosin, metoprolol and bromocriptine in additional models of IRD.

The research was noteworthy in that it established the utility of TMB combo therapy useful in several murine models and a canine model of IRD all with varying pathophysiology of disease.

Thank you for summarizing adequately.

1. In presenting the data regarding adrenergic and dopaminergic signaling in Rd10 mice (lines 242-244, Figure 1M), representation of results were unclear with current figure and text as to whether there were any significant differences between Veh treated and TMB treated Rd10 mice; it appears that comparisons were made between dark-reared mice and these two groups but not between veh and treatment group. Clarification and expanding upon this section of the results is especially important given the presumed mechanism in these catecholaminergic GPCR drugs used in this study, and the presumed affect on the pathology in these IRD models as touched upon in the discussion (lines 623-656).

Thank you for the relevant comment. The idea in Figure 1M and Figure 6P is to show gene expression changes in the catecholaminergic (CA) system in Rd10 and *Rpe65*^{-/-} mouse retinas in general (regardless of treatment). We do not propose that the efficacy of the TMB treatment comes via transcriptomic modulation of the CA system, but by decreasing signaling via CAergic postsynapses. This change in signaling does not necessarily lead to any changes in gene expression, as was seen in the Rd10 data. This is also mentioned in the results section: *“Finally, since the TMB cocktail consists of catecholaminergic drugs, we investigated changes in genes encoding catecholamine neurotransmitter receptors, as well as catecholamine neurotransmitter-synthesizing and -eliminating enzymes. Many of the catecholamine neurotransmitter signaling-related genes were significantly upregulated in rd10 mice regardless of TMB treatment (Fig. 1M).”*

We further elaborated the idea better and did some modifications into discussion. The relevant part now states: *“In our RNA-seq data, many genes related to the catecholaminergic system show disease progression-dependent dysregulation in rd10 mice; for example, Maa and Comt show upregulation. Although TMB treatment does not significantly alleviate this part of transcriptomic dysregulation, it does directly target the catecholaminergic signaling at receptor level.”*

For the reviewer’s interest, we show the regulation as scatter plots below. We did not add these figures in the manuscript, because similar data is in heatmaps.

Fig. 2. Genes of catecholaminergic receptors, and catecholamine synthetizing and metabolizing enzymes, from Rd10 retina bulk RNA-seq.

Fig. 3. Genes of catecholaminergic receptors, and catecholamine synthesizing and metabolizing enzymes, from *Rpe65*^{-/-} retina bulk RNA-seq.

2. In assessing changes in M and S(UV) cones in Rd10 treated mice (Fig 1U-Z), it appears that while M cones were improved, only one outlier mouse appears to have an increase in both S cone count and ERG. Given discussion of results on lines 255-260. Would recommend additional n's to clarify this finding.

We believe that sample size of 25 (12 vehicle, 13 TMB) is sufficient in the ERG analysis presented in Figs. 1W & X. We performed the D'Agostino & Pearson normality test for the data which showed that normality cannot be assumed. For this reason, using outlier tests is not advisable (such as the ROUT method or the Grubb's test). Therefore, we used nonparametric Mann-Whitney U-test. Even if we would manually remove this one high value from TMB dataset, it would not substantially change results or their interpretation. However, we investigated the data more closely and inspected single sweeps during the ERG recording. We also carefully inspected if UV cone counting could have mistakenly led to abnormally high counts. Both analyses were OK.

3. In evaluating changes in RPE65^{-/-} mice, there are presumed largely inner retinal changes given the ERG findings. It would have been helpful to also conduct similar scRNA-seq experiment to provide greater detail on changes occurring on which inner retinal cells, as was performed and discussed on dark-reared Rd10 mice.

We added data to the current manuscript. We ran an additional 5-week-long TMB trial in *Rpe65*^{-/-} mice and performed retinal single-cell RNA-seq. This data is now presented in Figs. 6P-R and S10-S12. Because part of the bulk-RNA-seq is less informative, we moved parts of it to the supplementary data (now Figure S9). We added the following discussion (lines 596-598): "*In Rpe65*^{-/-} mice, TMB treatment enhanced mitochondrial gene expression and metabolism-associated gene sets in rod, RBC and MG clusters, which may explain the improved visual function as detected by ERGs and OMR.". The relevant data is shown in Fig. S12.

4. Lastly, studies on the Pde6a^{-/-} canine model showed less of an effect, possibly largely due to administration and dosing/undertreatment. Assessing drug concentration levels in eyecup, even on WT or control dogs, would be helpful to elucidate whether this was the main limiting factor, or whether lack of effect may be due to either mechanistic/pathophysiology differences in a different IRD model or inherent differences in species and GPCR signaling.

This is a valid comment. However, even if we did drug level assessment from eyecups, retinas, and blood separately, we would not obtain definite answer whether the lack of effect was due to underdosing, or mechanistic/pathophysiology difference. Because of this uncertainty plus ethical concerns relating to the use of more dogs, we reasoned that addressing this issue with more experiments is not worthwhile. Instead, we added into the main text the possibility that the lack of effect could be due to pathophysiological differences between the models.

The added text in discussion (lines 644-649):

"Since M was most important for efficacy within our therapeutic strategy, we postulate that M underdosing is the best candidate for explaining why TMB's therapeutic effect was less apparent in PDE6A^{-/-} dogs than in Rd10 mice, although the difference could also be related to

pathophysiological differences between the models. Notably, the phenotype in PDE6A^{-/-} dogs is similar to rd10 mice but more severe as PDE6A^{-/-} dogs do not express any rod-specific PDE6 and completely lack rod function (61) whereas Rd10 mice initially demonstrate both (39)."

Other than the above recommendations on data interpretation and findings, data interpretation and methodology of the rest of the experiments presented were sound and contributes to the important search of additional, more practical therapeutics for IRDs.

Thank you for this encouraging verdict.

References cited in the letter:

- Barabas, Peter, Carolee Cutler Peck, and David Krizaj. 2010. "Do Calcium Channel Blockers Rescue Dying Photoreceptors in the Pde6b (Rd1) Mouse?" *Advances in Experimental Medicine and Biology* 664: 491–99. https://doi.org/10.1007/978-1-4419-1399-9_56.
- Chen, Yu, Grazyna Palczewska, Ikuo Masuho, Songqi Gao, Hui Jin, Zhiqian Dong, Linn Gieser, et al. 2016. "Synergistically Acting Agonists and Antagonists of G Protein-Coupled Receptors Prevent Photoreceptor Cell Degeneration." *Science Signaling* 9 (438): ra74. <https://doi.org/10.1126/scisignal.aag0245>.
- Kanan, Yogita, Mahmood Khan, Valeria E Lorenc, Da Long, Rishi Chadha, Jason Sciamanna, Ken Green, and Peter A Campochiaro. 2019. "Metipranolol Promotes Structure and Function of Retinal Photoreceptors in the Rd10 Mouse Model of Human Retinitis Pigmentosa." *Journal of Neurochemistry* 148 (2): 307–18. <https://doi.org/10.1111/jnc.14613>.
- Kern, Timothy S, Yunpeng Du, Jie Tang, Chieh Allen Lee, Haitao Liu, Alyssa Dreffs, Henri Leinonen, David A Antonetti, and Krzysztof Palczewski. 2021. "Regulation of Adrenergic, Serotonin, and Dopamine Receptors to Inhibit Diabetic Retinopathy: Monotherapies versus Combination Therapies." *Molecular Pharmacology* 100 (5): 470–79. <https://doi.org/10.1124/molpharm.121.000278>.
- Kilicarslan, Irem, Lucia Zanetti, Elena Novelli, Christoph Schwarzer, Enrica Strettoi, and Alexandra Koschak. 2021. "Knockout of CaV1.3 L-Type Calcium Channels in a Mouse Model of Retinitis Pigmentosa." *Scientific Reports* 11 (1): 15146. <https://doi.org/10.1038/s41598-021-94304-3>.
- Korstanje, Cees, Walter Krauwinkel, and Francisca L C van Doesum-Wolters. 2011. "Tamsulosin Shows a Higher Unbound Drug Fraction in Human Prostate than in Plasma: A Basis for Uroselectivity?" *British Journal of Clinical Pharmacology* 72 (2): 218–25. <https://doi.org/10.1111/j.1365-2125.2010.03870.x>.
- Pitkänen, Leena, Veli-Pekka Ranta, Hanna Moilanen, and Arto Urtti. 2007. "Binding of Betaxolol, Metoprolol and Oligonucleotides to Synthetic and Bovine Ocular Melanin, and Prediction of Drug Binding to Melanin in Human Choroid-Retinal Pigment Epithelium." *Pharmaceutical Research* 24 (11): 2063–70. <https://doi.org/10.1007/s11095-007-9342-0>.
- Shibagaki, Keiichi, Kazuyoshi Okamoto, Osamu Katsuta, and Masatsugu Nakamura. 2015. "Beneficial Protective Effect of Pramipexole on Light-Induced Retinal Damage in Mice." *Experimental Eye Research* 139 (October): 64–72. <https://doi.org/10.1016/j.exer.2015.07.007>.

REVIEWERS' COMMENTS

Reviewer #1 (Remarks to the Author):

All of my comments have been addressed adequately. While I regret that due to word count limitations not all issues could be addressed as deeply as I would have preferred, neither author nor I can challenge the word count limits.

Reviewer #2 (Remarks to the Author):

The authors satisfactorily addressed my (referee 2) concerns.

Reviewer #3 (Remarks to the Author):

The noteworthy significance of the study is of repurposing common GPCR agonist and antagonist medications and showing its effects on intracellular cAMP and Ca²⁺ signaling on various animal models of IRDs.

It is of significance to the field in its potential wide application across various IRDs in a gene-agnostic way.

The authors have addressed previous comments and provided additional evidence and data analysis, as well as made revisions on their interpretations and conclusions in this revision.

I would recommend acceptance of revised manuscript.